# Benchmark dataset and deep learning method for global tropical cyclone forecasting

Cheng Huang [1,5], Pan Mu[1,5], Jinglin Zhang[2], Sixian Chan[1], Shiqi Zhang[1], Hanting Yan[1], Shengyong Chen[3] & Cong Bai [1,4] ✉

Accurate tropical cyclone (TC) forecasting is critical for disaster prevention. While deep learning shows promise in weather prediction, existing approaches demonstrate limited accuracy in TC track and intensity forecasting, hindered by the lack of open multimodal datasets and insufficient integration of meteorological knowledge. Here we propose TropiCycloneNet containing $TCN_D$ - a open multimodal TC dataset spanning six major ocean basins with 70 years of multi-source data, and $TCN_M$ - an AI-meteorology integrated prediction model including multiple modules such as Generator Chooser Network and Environment-Time Net. Comprehensive evaluations demonstrate that $TCN_M$ outperforms both existing deep learning methods and official meteorological forecasts across multiple metrics. This advancement stems from synergistic optimization of our meteorologically-informed architecture and the dataset's comprehensive spatiotemporal coverage. The released resources and method can attract more researchers to the field, thereby accelerating data-driven tropical cyclone prediction research.

Tropical cyclones (TCs), referred to as typhoons, hurricanes, or cyclonic storms depending on the region, are intense and dynamic atmospheric systems. They typically originate in tropical to temperate latitudes and play a crucial role in regulating the global distribution of heat and momentum[1]. Additionally, powerful TCs can have potentially catastrophic impacts on maritime vessels and offshore platforms. Moreover, when these TCs make landfall, they often trigger a series of destructive natural disasters, including gales, storm surges, and floods[2], resulting in significant economic losses and posing threats to human lives. With climate change, the intensity and destructiveness of tropical cyclones have shown an increasing trend over the past few decades[3,4]. To mitigate these disasters, accurate advance prediction of the track and intensity of TCs is essential. Track and intensity predictions are the most crucial aspects of TC prediction, and are also the focus of our forecasting efforts in this study. Despite significant research efforts, tropical cyclone (TC) forecasting remains a major

scientific challenge[5]. This difficulty arises from the multitude of influencing factors, such as large-scale atmospheric circulation, the positioning of the subtropical high, the internal pressure structure and center of the TC, sea surface temperature (SST), and other environmental conditions[6], as illustrated in Fig. 1. The interactions among these factors are complex, and their combined effects on TC evolution are not yet fully understood. Given the critical need for accurate TC forecasts, the scientific community has been actively investigating this topic since the early 20th century[7]. Over time, researchers have analyzed TC behavior and explored some prediction strategies, including empirical, statistical, and numerical modeling techniques[8,9].

Numerical Weather Prediction has become the dominant technique for forecasting TCs, widely utilized by national meteorological centers such as the Chinese Central Meteorological Observatory (CMO). These systems rely on high-performance computing to model the complex atmospheric processes involved in TC evolution.

[1]College of Computer Science, Zhejiang University of Technology, Hangzhou, China. [2]School of Control Science and Engineering, Shangdong University, Jinan, China. [3]School of Computer Science and Engineering, Tianjin University of Technology, Tianjin, China. [4]Zhejiang Key Laboratory of Visual Information Intelligent Processing, Hangzhou, China. [5]These authors contributed equally: Cheng Huang, Pan Mu. ✉e-mail: congbai@zjut.edu.cn

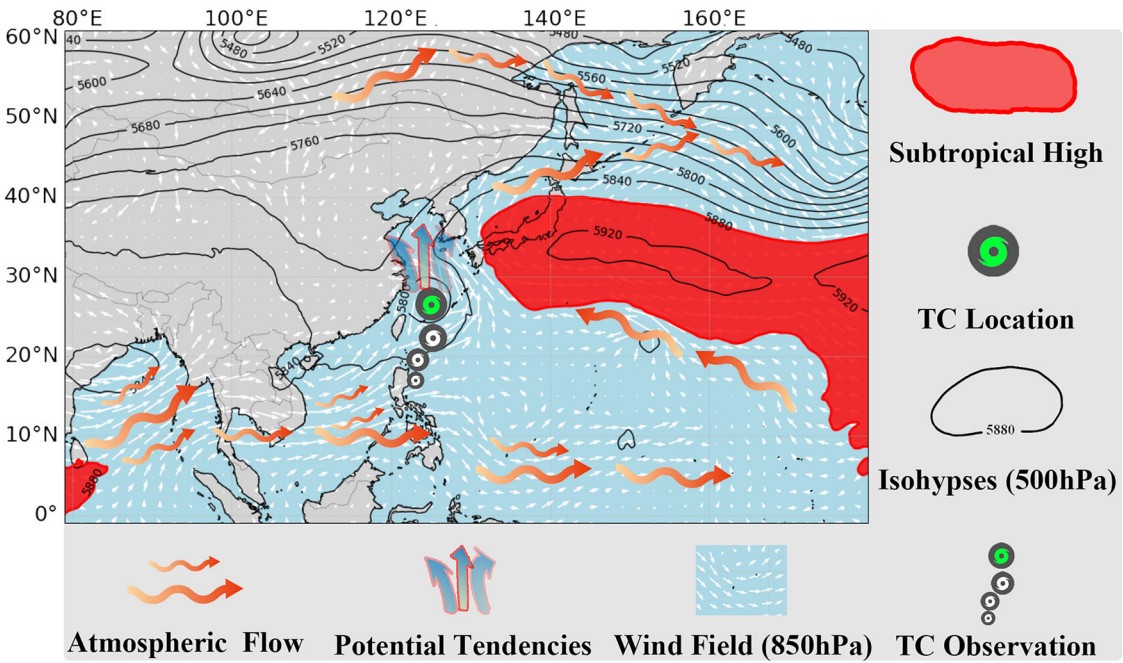

**Fig. 1 | The development of a tropical cyclone with environment factors.** The development of a tropical cyclone is influenced by various environmental factors, such as the subtropical high and atmospheric circulation. These factors can all affect the future evolution and track of the tropical cyclone. Best viewed in color.

However, the computational expense and resource of running such simulations on supercomputers remain significant challenges[10]. In recent years, deep learning has emerged as a promising alternative, capable of managing complex prediction tasks using relatively computational resource, such as a few graphics processing units (GPUs). Deep learning has seen widespread success in domains such as computer vision[11], time series forecasting[12], and a variety of interdisciplinary applications[13,14]. Leveraging the growing availability of diverse meteorological datasets, deep learning models—especially Recurrent Neural Networks (RNNs)[15]—offer strong potential for advancing TC prediction. The foundation of applying deep learning technology in specific research areas lies in the availability of high-quality datasets. A high-quality dataset can not only lower the threshold for people to access the training data but also fosters the development of the relevant research field. A prime example is the ImageNet dataset[16], which was created in 2010 for image classification tasks. In 2012, the deep learning model AlexNet[17] demonstrated its potential by achieving remarkable performance on this dataset, outperforming previous methods and significantly enhancing accuracy. Comparing to previous methods, AlexNet dramatically improved the accuracy of image classification task on ImageNet. However, it still fell short of outperforming than human experts in this task. Just three years later, the development of deep learning technology led to the creation of ResNet[18], which easily defeated the human experts on image classification task on ImageNet. This rapid progress in computer vision can be attributed to the availability of ImageNet as an open dataset, which played a crucial role in the success of deep learning. It is no doubt that behind the success of deep learning for computer vision, contributions of the open dataset ImagNet could not go unnoticed. Additionally, numerous exceptional datasets[19,20] released over the past decade have propelled development across various fields.

In the field of TC prediction, there are several important datasets, such as International Best Track Archive for Climate Stewardship (IBTrACS)[21], Tropical cyclones Dataset based on satellite cloud images (TCTSCI)[22], Satellite Image Dataset for Tropical Cyclones (Digital Typhoon)[23], and China Meteorological Administration Tropical Cyclone Best Track Data (CMA-BST)[24,25]. CMA-BST includes only

inherent attribute data, such as the track and intensity of TCs. Compared to CMA-BST, TCTSCI and Digital Typhoon contains meteorological grid data, such as satellite images and reanalysis data; however, both datasets are limited to TCs in the Western North Pacific region. In contrast, IBTrACS is a global best track dataset for TCs, but it only includes inherent attribute data. Furthermore, these datasets neglect some important environmental factors of TCs, which play a critical role in TCs' development. Thus, these datasets do not provide sufficient information for deep learning models to effectively learn and predict TC. That is, there is still no open large-scale, multi-modal dataset for TC prediction, which hinders its development. Although some researchers have realized the potential of deep learning for TC forecasting, they lack an appropriate benchmark dataset to design and evaluate their deep learning models. As a result, they are forced to evaluate their models using small-scale or single-modal TC datasets[21,24]. Besides, different models are evaluated on varied test set, which makes it challenging to compare their performance directly. It means that researchers face difficulties in fairly evaluating various models. This situation hampers the development of deep learning in TC forecasting. What is more, many Artificial Intelligence (AI) researchers are eager to engage in AI for Science research, such as TC forecasting, and contribute to the field. However, they first must tackle the challenge of collecting training data, as they often lack the time and specialized knowledge required for gathering and processing data, such as meteorological data. They tend to gravitate towards fields with accessible open datasets. Thus, to attract much participation of researchers and leverage AI to advance TC prediction, establishing an open, large-scale, multi-modal, and high-quality TC dataset is essential.

Although facing a shortage of open and suitable TC datasets, researchers have made notable attempts to utilize deep learning methods for TC prediction. Initially, single-modal data is used as the input of an RNN to predict TCs[26,27]. Recognizing the limitations of relying solely on single-modal data for comprehensive TC representation, researchers have explored integrating heterogeneous meteorological data in various TC prediction methods[28,29]. Researchers are exploring new paths in tackling TC prediction challenges, which is highly significant. However, in their proposed methods, most

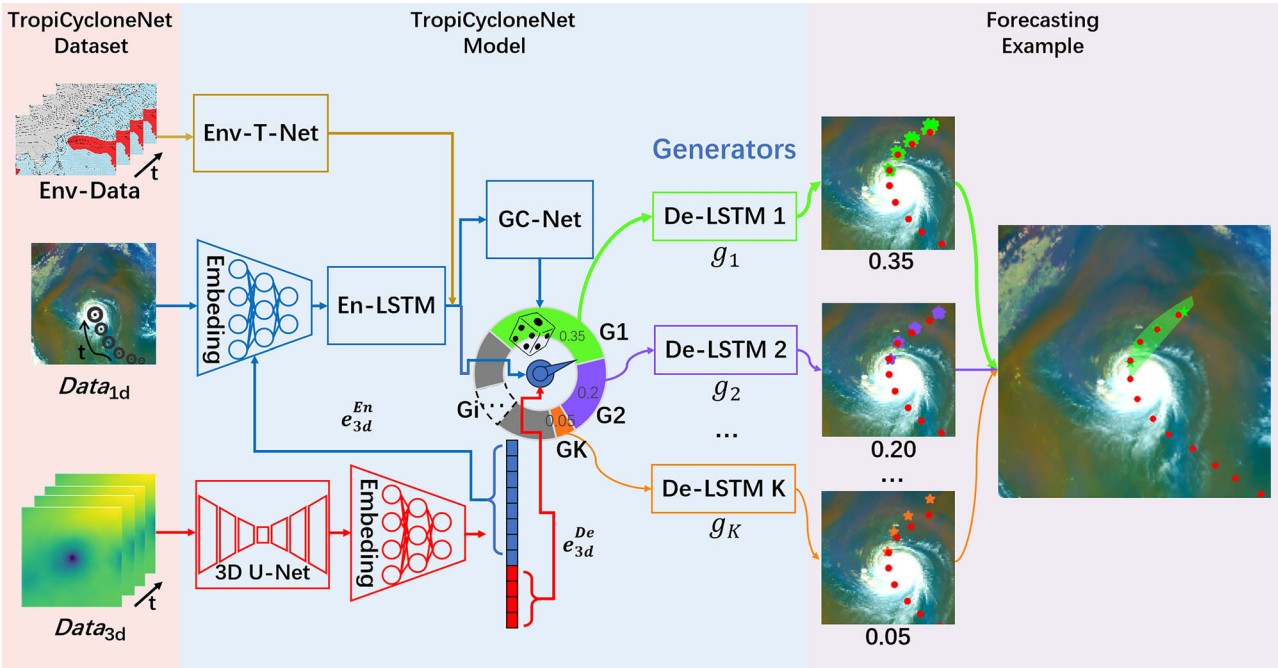

**Fig. 2 | TropiCycloneNet framework.** It includes TropiCycloneNet Dataset (TCN$_D$) and TropiCycloneNet Model (TCN$_M$). TCN$_D$ includes various data with time series. For TCN$_M$, the golden branch is the environment data (Env-Data) encoder, called Environment-Time Network (Env-T-Net). The blue branch is the inherent attributes data of tropical cyclone (Data$_{1d}$) encoder, called 1D-Data Encoder. The red branch is the meteorological grid data (Data$_{3d}$) encoder, called 3D-Data Encoder. The Roulette selects different generators by the probability array, the circular ring below GC-Net, from Generator Chooser Net. All predictions from selected generators constitute the multiple potential tendencies of tropical cyclone. Best viewed in color.

researchers thought in terms of the AI perspective during designing the model, often neglecting the meteorological perspective of the task. They attempted to address the complex, interdisciplinary challenge of TC forecasting solely with AI knowledge. Overall, there are some drawbacks to previous methods, like (1) due to using small-scale training TC data, these methods may underperform in predicting TCs from sea areas that are not included in their training data. It means that a model might get good results in a specific sea area, like the Western North Pacific, but falter in others, because this model does not "see" the TCs from other regions and thus has not learned their development patterns. (2) Previous deep learning methods often overlook the impact of environment-related factors, while these factors are critical in understanding the development of TC[30]. (3) Most of the deep learning methods[26,31,32] predict only a single future tendency of TC, while most meteorologists generally prefer methods employing a mechanism of predicting multiple potential TC tendencies, as illustrated by the semitransparent green region at the left of Fig. 2. These shortages prevent previous methods from achieving satisfactory performance in TC prediction.

The emergence of several large-scale weather models, like Pangu-Weather[32], Fengwu[33], and Graphcast[34], has brought various meteorological forecasting issues to the attention of scientists from diverse fields. These general meteorological models have achieved breakthroughs in medium-range global weather forecasting, greatly aiding deep learning methods in various downstream meteorological tasks, such as TC forecasting. To fully leverage the capabilities of these large-scale models, it is urgent to build an open high-quality dataset that enables researchers to fine-tune these models for specific downstream meteorological tasks. This will facilitate better outcomes in their respective tasks. Besides, the cost of training these large-scale models with nearly 100 A100 GPUs for about a month[32] is prohibitive for most research and development teams. Consequently, the development of lightweight models is essential to reduce the computing

resource requirements for researchers, thereby encouraging broader participation.

To address these problems, we propose solutions in two key areas: the TC dataset and a dual perspective deep learning method integrating meteorology and AI. Thus, in this work, we propose the TropiCycloneNet including a benchmark dataset−TropiCycloneNet Dataset and an innovative TC forecasting method - TropiCycloneNet Model. The main contributions of this work include: (1) the TropiCycloneNet Dataset (TCN$_D$) is an open large-scale multi-modal TC dataset. The TCN$_D$ includes near 70 years TC data over 6 main sea areas including North Atlantic (NA), Eastern North Pacific (EP), Western North Pacific (WP), North Indian (NI), South Indian (SI), and South Pacific (SP), with total of 3630 TCs. The TCN$_D$ not only includes the inherent attribute data of TC called Data$_{1d}$ (e.g., longitude, latitude, and wind) but also the meteorological grid data called Data$_{3d}$ (e.g., meteorological fields). Additionally, a significant amount of TC environmental data has been collected to enrich the dataset for deep learning models. (2) We design the Environment-Time Network (Env-Time-Net), one of the important modules in our generic, extensible, multi-modal TC prediction model−TropiCycloneNet Model (TCN$_M$) that can be trained on only one GPU for a day. Env-Time-Net improves TC prediction performance by incorporating the temporal information of environment that is traditionally overlooked but is very important. To our knowledge, this is an attempt to develop an environment-focused module in TC deep learning prediction methods. (3) To enhance our model's ability to predict multiple potential TC tendencies, we design the Multi-generator with Generator Chooser Network (GC-Net) to tackle the prediction of undesired out-of-distribution (OOD) samples and the insufficient learning ability of previous TC prediction methods. (4) Extensive experiments have been conducted on the TCN$_D$. Our model achieved state-of-the-art performance in deep learning and surpassed the results of some official meteorological agencies in most metrics.

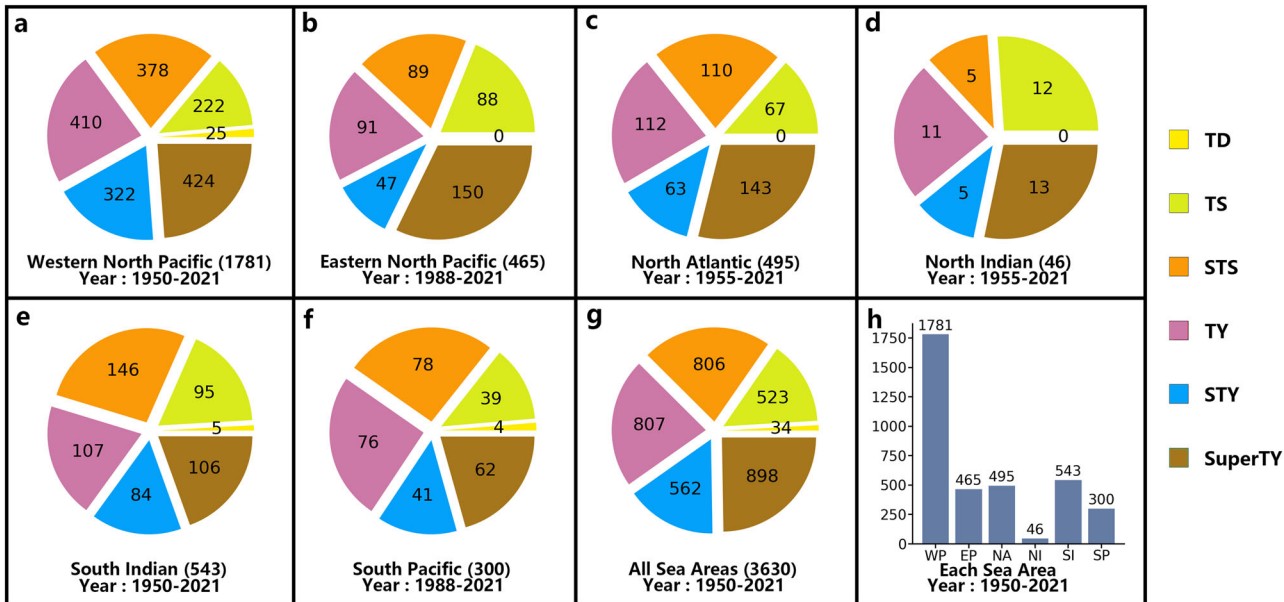

**Fig. 3 | TropiCycloneNet Dataset overview.** The distribution of tropical cyclones (TCs) with varying intensities in TropiCycloneNet Dataset (TCN$_D$) across different sea areas, categorized as tropical depression (TD, 10.8–17.1 m/s), tropical storm (TS, 17.2–24.4 m/s), severe tropical storm (STS, 24.5–32.6 m/s), typhoon (TY, 32.7–41.4 m/s), strong typhoon (STY, 41.5–50.9 m/s), and super typhoon (SuperTY, >51.0 m/s). Sub-figures **a**–**f** illustrate the count of TCs with different intensities in specific areas. For example, in sub-figure a, the number 378 indicates there are 378 severe tropical storm intensity TCs in the Western North Pacific region in TCN$_D$. The number 1781 represents the total TCs of the Western North Pacific in TCN$_D$. Sub-figure **g** presents the count of TCs of various intensities across these six sea areas, while sub-figure **h** displays the total count of TCs in each sea area. Best viewed in color.

## Results

### Definition of the problem

We consider TC track and intensity prediction as a spatiotemporal prediction problem. Thus, we build TCN$_D$, a multi-modal dataset with time information. The TCN$_D$ data is divided into three parts: inherent attribute data, meteorological grid data, and environment data. The inherent attribute data, such as longitude, latitude, pressure, and wind, are denoted as $Data_{1d}$. We denote the meteorological grid data, such as geopotential height[35], as $Data_{3d}$. We refer to environment data as Env-Data, which includes the month, the velocity of movement, history of moving direction (24 h), and the region of the subtropical high. Assuming a TC developing in the Pacific Ocean, we receive the observation data $x_t$ of this TC as model input denoted as $X = \{x_1, x_2, \ldots, x_n\}$. The data for each time point is represented as $x_t = \{x_{1d}^t, x_{3d}^t, x_{env}^t\}$, where $t \in \{1, 2, \ldots, n\}$, $n$ is the sequence length of the input data, $x_{1d}^t \in Data_{1d}$, $x_{3d}^t \in Data_{3d}$, and $x_{env}^t \in$ Env-Data. When we obtain the input $X$, our goal is for TCN$_M$ to output $\hat{Y} = \{\hat{y}_{n+1}, \hat{y}_{n+2}, \ldots, \hat{y}_{n+m}\}$, which closely approximates the actual future track and intensity of TC (ground-truth). The actual future tendency of the TC is denoted as $Y = \{y_{n+1}, y_{n+2}, \ldots, y_{n+m}\}$. In addition, the real and prediction data of TC on each point time are defined as $y_t = \{y_{track}^t, y_{intensity}^t\}$ and $\hat{y}_t = \{\hat{y}_{track}^t, \hat{y}_{intensity}^t\}$ respectively, where $t \in \{n+1, n+2, \ldots, n+m\}$. Here, $m$ is the sequence length of the prediction data of TC. In this work, we set $n = 8$ and $m = 4$, which means that we input 42 h historical TC data into the model and the model outputs predictions for the TC track and intensity over the next 24 h.

### TropiCycloneNet Dataset

The TropiCycloneNet Dataset (TCN$_D$) is a large-scale TC prediction dataset, including the majority of TCs that have developed in six main sea areas in recent decades. According to Fig. 3, the dataset includes 3630 collected TCs across six intensity categories. The majority of these TCs originated in the Southern Hemisphere. In addition, TCN$_D$ is a multi-modal dataset with rich heterogeneous meteorological data. To provide a clear overview of our dataset and

inform readers about its contents, TCN$_D$ is divided into inherent attribute data, meteorological grid data, and environment data. The details of meteorological grid data, inherent attribute data, and environment data are shown in Table 1.1, 1.2, and 1.3 respectively. The contributions of each data for helping deep-learning models to understanding TCs are shown in Table 2. The primary differences between TCN$_D$ and other well-known TC datasets (e.i. CMA-BST[24], TCTSCI[22], Digital Typhoon[23] and IBTrACS[21]) are detailed in Table 3. It took a lot of effort to collect and align these different types of data. We also collaborated with renowned meteorological experts and invite them to review our dataset to improve the quality of TCN$_D$. TCN$_D$ is also an open dataset. Furthermore, we will update and maintain our dataset annually to ensure users have access to the latest TC-related data. We have released it on Github and Zenodo to build a bridge between TC prediction and AI, encouraging more researchers to engage in this field. Each component of TCN$_D$ will be detailed in the following sections.

**Inherent attributes data.** Inherent attribute data constitutes the foundational data in TCN$_D$, directly describing the location and intensity of TCs. It includes the data of longitude, latitude, pressure, and wind at or near the TC's center, collected from official TC records[21,24], which is shown in Table 1.2. Additionally, these attributes serve as the target variables for prediction in our study. The inherent attribute data are one-dimensional data (scalar variables with time series), thus referred to $Data_{1d}$.

The data in $Data_{1d}$ was normalized by some rules because normalized data is suitable for deep learning methods to extract useful information. All the rules are shown as follows:

The data in $Data_{1d}$ were normalized according to certain rules, as normalized data are more suitable for deep learning methods to extract useful information. In the ID column, a value of 0.0 indicates that this record is the first record of this TC, and the time between 0.0 and 1.0 represents a 6-h interval. In the LONG column, these values represent the normalized longitude of the TC. The rule for normalizing

## Table 1 | Details of TropiCycloneNet Dataset

| Data Name | Sea surface temperature | Geopotential height | U-component of wind | V-component of wind |
|---|---|---|---|---|
| Spatial resolution | 0.25° | 0.25° | 0.25° | 0.25° |
| Time resolution | 6 hourly | 6 hourly | 6 hourly | 6 hourly |
| Pressure Level (hPa) | / | 200, 500, 850, 925 | 200, 500, 850, 925 | 200, 500, 850, 925 |
| Size | 81*81 | 81*81 | 81*81 | 81*81 |
| (around the TC center) | (20°*20°) | (20°*20°) | (20°*20°) | (20°*20°) |

| ID | LONG | LAT | PRES | WND | YYYYMMDDHH | Name |
|---|---|---|---|---|---|---|
| 6 | −10.42 | 2.68 | 0.8 | −1 | 2019090112 | LINGLING |
| 7 | −10.56 | 2.86 | 0.8 | −1 | 2019090118 | LINGLING |
| 8 | −10.8 | 3.06 | 0.76 | −0.88 | 2019090200 | LINGLING |
| 9 | −11.02 | 3.36 | 0.76 | −0.88 | 2019090206 | LINGLING |
| 10 | −11.14 | 3.64 | 0.7 | −0.8 | 2019090212 | LINGLING |
| 11 | −11.2 | 3.78 | 0.6 | −0.68 | 2019090218 | LINGLING |
| 12 | −11.22 | 3.92 | 0.5 | −0.6 | 2019090300 | LINGLING |

| Key | Value |
|---|---|
| Moving Velocity | 0.200337442 |
| Month | 000000001000 |
| Location Longitude | 0*12 1 0*23 |
| Location Latitude | 00000000100 |
| History Direction (24 h) | 00000010 |
| History Intensity Change (24 h) | 0010 |
| Subtropical High | GPH |

[1]Details of $Data_{3d}$: We crop data of 81*81 size from the fifth-generation atmospheric reanalysis of the global climate (ERA5) data to ensure that these data cover the 20°*20° region around the tropical cyclone (TC) center.

[2]Examples of $Data_{1d}$: ID is the time step of TC, LONG is the Longitude (0. 1°E) of TC, LAT is the Latitude (0. 1°N) of TC, PRES is the minimum pressure (hPa) near the TC center, WND is the 2-min mean maximum sustained wind (MSW; m/s) near the TC center. YYYYMMDDHH is the date and time in UTC of TC, Name is the name of the TC.

[3]Environment data: The environment data of TC LINGLING at the date 2019/09/06 18:00. 0*12 1 0*23 means the one-hot code includes 12 "0"s, a "1", and 23 "0"s.

the longitude is as follows:

$$LONG_{normalized} = \frac{LONG - 1800}{50} \qquad (1)$$

where LONG represents the original longitude data (in increments of 0. 1°E).

In the LAT column, these values represent the normalized latitude of the TC. The rule for normalizing the latitude is as follows:

$$LAT_{normalized} = \frac{LAT}{50} \qquad (2)$$

where LAT represents the original latitude data (in increments of 0. 1°E).

In the PRES column, these values represent the normalized pressure of the TC. The rule for normalizing the pressure is as follows:

$$PRES_{normalized} = \frac{PRES - 960}{50} \qquad (3)$$

where PRES represents the original pressure data (in hPa).

In the WND column, these values represent the normalized wind of the TC. The rule for normalizing the wind is as follows:

$$WND_{normalized} = \frac{WND - 40}{25} \qquad (4)$$

where WND represents the original wind data (in m/s).

**Meteorological grid data.** There are many meteorological grid data about TC, like various variables in ERA5 dataset[35]. By leveraging this

data, we can extract rich features to better represent the TC. For example, we can use the 500 hPa geopotential height data to describe the TC's pressure structure. The meteorological grid data are three-dimensional data (two-dimensional grid data with time series), hence referred to $Data_{3d}$. In addition, $Data_{3d}$ encompasses not only geopotential height but also other grid data, like SST, U-component of wind, and V-component of wind. All $Data_{3d}$ used in this study were collected from ERA5 reanalysis dataset. The details of $Data_{3d}$ are shown in Table 1.1.

After obtaining the $Data_{3d}$ data, we align them with $Data_{1d}$ based on the timestamp. We crop the data to cover 20°*20° region around the TC center. The spatial resolution is 0.25° and the temporal resolution is 6 h. We collect geopotential height, U-component of wind, V-component of wind at 200, 500, 850 and 925 hPa pressure levels. SST data is also part of our $Data_{3d}$ set. Although there are some TC-related variables in our dataset, the TC is too complex to be fully represented by these data. Therefore, we will continue to update variables to meet various requirements of users and improve the dataset collaboratively.

**Environment data.** Environment factors, such as subtropical height, and seasonality, play a critical role in TC development, influencing the tendency of both track and intensity. Therefore, to enable deep learning models to leverage environmental information and enhance prediction accuracy, we collaborated with meteorologists to identify and extract crucial environmental data from both $Data_{1d}$ and $Data_{3d}$, which is shown in Table 1.3. This environment data, referred to as Env-Data, includes the movement velocity, month, the relative location on the earth (Location Longitude and Location Latitude), 24-h history of moving direction, 24-h history of intensity change, and subtropical high region. Some of these factors are often overlooked by previous deep learning methods.

**Table 2 | Data's contribution for tropical cyclone forecasting**

| Data Type | Data Name | Contribution for TC forecasting |
|---|---|---|
| $Data_{1d}$ | LONG | The target attribution to be predicted |
| | LAT | The target attribution to be predicted |
| | PRES | The target attribution to be predicted |
| | WND | The target attribution to be predicted |
| $Data_{3d}$ | Sea surface temperature | TCs mainly intensify by absorbing heat energy from the ocean. Warm sea surface temperatures (typically above 26.5°C) provide the energy that drives the development and strengthening of typhoons. High-temperature ocean regions promote water vapor evaporation, increasing the moisture supply to the TC and further enhancing its strength. |
| | Geopotential Height | As a powerful low-pressure system, the strength of a TC is closely related to the surrounding geopotential height field. Strong TC exhibit lower geopotential heights at various atmospheric levels, where this is more pronounced. By monitoring the geopotential heights at these levels, the intensity trend of a TC can be assessed. |
| | U-component of wind and V-component of wind | The U-component (zonal wind) and V-component (meridional wind) together describe the horizontal wind structure in a three-dimensional wind field. By analyzing the UV components, the wind speed and direction around a TC can be accurately characterized, providing insight into the TC's convective structure, airflow patterns, and surrounding environment. This is crucial for assessing the intensity and wind field variations of the TC. The TC's path is typically guided by large-scale atmospheric circulation, and these steering currents can be captured through the UV components. |
| Env-Data | Moving Velocity | The moving velocity of a TC directly influences the prediction of its future path. By understanding its current speed, it is possible to predict the geographical location it will reach at a given time. The moving velocity of a TC is closely related to changes in its intensity. Generally, slower-moving TCs remain over warm ocean areas for longer periods, providing more opportunity to absorb energy, which may lead to intensification. In contrast, faster-moving TCs may weaken more quickly when interacting with cold air or land. Therefore, the moving velocity helps in estimating the intensity change trends of a TC. |
| | Month | The occurrence of TCs follows a distinct seasonal pattern, typically being most active during the summer and autumn months. By understanding the month, it is possible to better predict the likelihood and frequency of TC formation. Seasonal changes in atmospheric circulation during different months can lead to variations in TC track patterns. For instance, in the Northwest Pacific, when the subtropical high is strong during the summer, TCs may move westward, affecting East Asia. However, in autumn, as the subtropical high shifts eastward, TCs are more likely to move northward or northeastward. |
| | Location Longitude and Location Latitude | Different oceanic and atmospheric conditions across regions influence TC formation and development. Warm areas like the Northwest Pacific and South China Sea favor typhoon formation, while colder, higher-latitude regions are less conducive. Each region shows distinct patterns in TC activity, and analyzing historical data helps improve predictions by revealing trends in path, frequency, and intensity. |
| | History Direction (24 h) | By analyzing the TC's movement direction over the past 24 h, the model can better assess the future track trends of the TC. TC movement direction typically does not undergo drastic changes in a short period, making historical direction a reliable basis for predicting future paths. The TC's movement is guided by the surrounding atmospheric circulation, such as the position of the subtropical high or troughs. Changes in the historical direction reflect the influence of these steering flows, helping to predict whether the TC will deviate towards a particular direction. |
| | History Intensity Change (24 h) | By analyzing the TCs intensity change over the past 24 h, the model can help determine the future trend of intensity changes. If the TC has rapidly intensified or weakened in the past 24 h, it may indicate that this trend will continue, unless significant changes occur in the external environment. The historical intensity change reflects the influence of the surrounding environmental conditions, such as sea surface temperature and wind shear. For example, if a TC has rapidly intensified over the past 24 h, it may suggest that it has passed through an area with high sea surface temperatures or low wind shear, and this information can help assess future intensity changes. |
| | Subtropical High | The subtropical high, a large high-pressure system typically located between 20° and 30° latitude, is a key factor influencing the path of tropical cyclones (TCs). TCs often follow the edge of the subtropical high, guided by its steering flow: a stronger high tends to steer TCs westward, while a weaker or eastward-shifted high allows for northward or northeastward movement. Although its main influence is on TC tracks, the subtropical high can also affect intensity by providing a stable environment with low wind shear, which helps maintain or enhance TC strength. |

Here, we also process the values of each variable in Env-Data to make them more suitable as input for deep learning models. The subsection Explanation of Environment Data in section "Data Availability" explains how we process the values. Specifically, we use the one-hot encoding of Location Longitude and Latitude to represent the TC's relative location on Earth, which differs from the location data in $Data_{1d}$. We aim to eliminate the numerical magnitude relationship inherent in latitude and longitude through this encoding method, as such relationships have no physical significance, particularly for longitude. This encoding approach allows the model to focus more on the relative position of the TC on Earth. Different relative positions correspond to distinct environmental conditions, allowing the model to learn the differences between them and, consequently, make more accurate predictions.

**Table 3 | A comparison of TropiCycloneNet Dataset and other well-known tropical cyclone dataset**

| Contents Dataset | CMA-BST | TCTSCI | IBTrACS | Digital Typhoon | $TCN_D$ |
|---|---|---|---|---|---|
| $Data_{1d}$ | ✓ | ✓ | ✓ | ✓ | ✓ |
| $Data_{3d}$ | ✗ | ✓ | ✗ | ✓ | ✓ |
| Env-Data | ✗ | ✗ | ✗ | ✗ | ✓ |
| Global | ✗ | ✗ | ✓ | ✗ | ✓ |

$Data_{1d}$ represents inherent attributes data. $Data_{3d}$ refers to meteorological grid data. Env-Data means environment data. Global indicates whether the dataset encompasses nearly all sea areas on this planet.

As demonstrated in Table 1.3, the value of Moving Velocity 0.200337442 is a normalized value. The Month contains the season information of TC, as TCs developed over different sea area follow different development rules in different seasons. Therefore, Month is also important environment data. The value of the Month is represented as a one-hot code with a length of 12. The 12 months are divided into 12 categories. The 000000001000 indicates that the month of TC is September, as the ninth position of the one-hot encoded Month is 1. Additionally, the behavior of TCs over land and sea differs. The location of the TC is a critical attribute, as it indicates whether the typhoon is over land or sea. The values of Location Longitude and Location Latitude are also represented as one-hot codes. The area $(60°S–60°N, 0°E–360°W)$ is divided into a $12 \times 36$ grid. The Location Latitude 000000001000 and Location Longitude 0*12 1 0*23 indicate that TC is over the region from 30°N to 40°N and 120°E to 130°E. Historical data of TC also contain useful information. The track directions are divided into 8 categories: east, southeast, south, southwest, west, northwest, north, and northeast. What is more, the intensity change tendencies are classified into 4 categories: strengthening, strengthening and then weakening, weakening, and maintaining the same. The History Direction (24 h) 00000010 indicates the that the direction of movement in the last 24 h is north, while the History Intensity Change (24 h) 0010 indicates that the intensity of the TC has weakened in the last 24 h. Otherwise, we extract more environment information from $Data_{3d}$. The TC typically moves along the edge of the subtropical high. Geopotential Height (GPH) contains the location and extent information of subtropical high. Therefore, we add 500 hPa GPH data to our environmental dataset. Although these data cannot fully represent all environmental factors of the TC, it is still an interesting attempt to build a dataset of environmental factors for TCs. We will continue to refine this dataset and collect more valuable data.

## TropiCycloneNet model
To take full advantage of our dataset $TCN_D$, TropiCycloneNet Model $(TCN_M)$ is designed. This GAN-based TC forecasting model comprises several key modules: 3D-Data Encoder, 1D-Data Encoder, Environment-Time Network, Multiple Generators, Generator Chooser Network, and Discriminator. The primary framework of $TCN_M$ is shown in Fig. 2. The 3D-Data Encoder (red branch) and 1D-Data Encoder (blue branch) are based on 3D-Unet[36] and Long Short-Term Memory (LSTM), respectively. They are designed to extract crucial spatiotemporal information from diverse data modalities and dimensions for a more comprehensive TC representation. Additionally, significant attention is given to the impact of environmental factors. Thus, the $TCN_M$ incorporates the Environment-Time Network (Env-T-Net) represented by the gold branch in Fig. 2. The Env-T-Net is specifically designed using Convolutional Neural Networks[37] and Transformer[38]. This network module models the TC's current environment and its evolution over recent days, enhancing the model's ability to understand and predict TC behavior. Given the environment's crucial role in TC behavior, capturing its temporal changes is imperative for our model. To enable more accurate multiple potential predictions, we incorporate multiple generators and the Generator Chooser Network (GC-Net). The Discriminator evaluates both real and synthetic TC data generated by $TCN_M$, enhancing the realism of our TC predictions.

## Experiments setup
Various experiments are conducted to demonstrate the significance of $TCN_D$ and the effectiveness of $TCN_M$. Extensive meteorological data spanning from 1950 to 2021 were collected as the training foundation for $TCN_M$, including longitude, latitude, pressure, and wind at or near the TC's center in $Data_{1d}$, 500 hPa geopotential height data in $Data_{3d}$, and all variables in Env-Data. From the data collected between 1950 and 2016, 80% was allocated to the training set, and 20% to the validation set. The data from 2017 to 2021 were designated as the test set.

## Metrics
Absolute error was used to evaluate the performance of the different methods in all experiments. For track prediction, the absolute distance error (km) between the actual and predicted trajectories was calculated. For intensity prediction, which includes pressure (hPa) and wind speed (m/s) at or near the TC's center, the absolute error between the actual and predicted data was also calculated.

## The performance of TropiCycloneNet model in different sea areas
To assess the performance of $TCN_M$, it was tested across six major sea areas worldwide. Additionally, we compared the performance of $TCN_M$ trained on all the sea areas ($TCN_M$) to that trained specifically on Western Pacific ($TCN_M^{WP}$), to demonstrate the significance of building a large-scale TC dataset. The Western Pacific (WP) region was chosen for comparison due to its high frequency of TC occurrences and the abundance of TC data. The results are presented in Fig. 4, revealing several interesting phenomena.

(1) It is no doubt that a large-scale dataset is of great importance for training a deep learning model. On one hand, $TCN_M$ consistently outperforms $TCN_M^{WP}$ across nearly all metrics. Particularly for predictions in the Southern Hemisphere (SI and SP regions), $TCN_M$ significantly outperforms $TCN_M^{WP}$, achieving 61.3–72.6% improvements in track prediction and 2.8–25.4% improvements in intensity prediction. It is noteworthy that $TCN_M^{WP}$ only learns the TC development patterns in the Northern Hemisphere, which explains its poor performance when encountering TCs from the Southern Hemisphere. In contrast, $TCN_M$ can tackle TCs from any sea area more easily than $TCN_M^{WP}$ because it has seen various TCs generated in both the Southern and Northern Hemisphere through $TCN_D$. On the other hand, comparing the performance of $TCN_M$ and $TCN_M^{WP}$ in the WP region reveals that models trained on more data outperform those trained on less. There are many TCs in other sea areas in Northern Hemisphere, which are similar to TCs generated in WP. This data enables deep learning models to encounter a greater volume of similar TCs, thereby improving their understanding of TC patterns. Overall, $TCN_D$ enables $TCN_M$ to achieve ~41.2% improvement in track prediction and 13.3% improvement in intensity prediction.

(2) $TCN_M$ demonstrates impressive performance in both track and intensity prediction across most sea areas, especially in the EP region. Regarding its performance in the SP region, there is a noticeable disparity compared to other sea areas. This phenomenon may be attributed to two main factors: the scarcity of training data in the Southern Hemisphere and the complex sea environment in the SP region where there are many islands. Nevertheless, based on these experimental results, it is evident that the integration of the extensive TC dataset $TCN_D$ is crucial, and $TCN_M$ exhibits great potential as a deep learning method.

## Comparison with state-of-the-art deep learning methods
First, we conducted experiments comparing $TCN_M$ to other state-of-the-art methods on the $TCN_D$. Then, we analyzed the experimental results from various perspectives. The results indicated that our method achieved the highest prediction accuracy among the deep learning methods tested.

**Deep learning methods for TC forecasting.** To comprehensively compare $TCN_M$ with other deep learning methods, we selected nine methods, categorized into three groups for comparison. The first group consists of classic time-series prediction methods, such as LSTM[15], GRU[39], and SGAN[40]. The second group includes modifications of time-series networks tailored for tropical cyclone (TC) prediction,

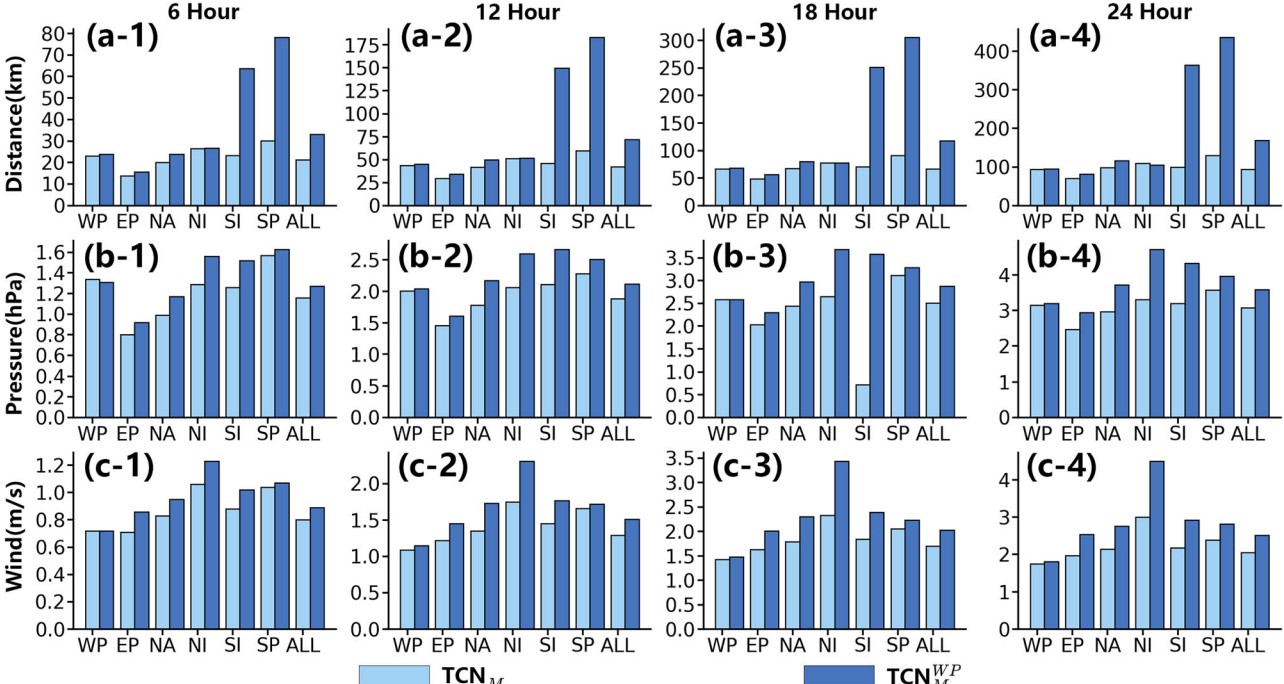

**Fig. 4 | The performance of TropiCycloneNet Model in different sea areas.** The sea areas include Western North Pacific (WP), Eastern North Pacific (EP), North Atlantic (NA), North Indian (NI), South Indian (SI), and South Pacific (SP). ALL means the average errors of all the six sea areas. Sub-figure (**a-1, 2, 3, 4**) show the performance of our model on tropical cyclone (TC) track prediction. Sub-figure (**b-1, 2,** 3, 4) and (**c-1, 2, 3, 4**) show the performance of our model on TC intensity prediction. The TropiCycloneNet Model ($TCN_M$) means that our model is trained on all the six sea areas. The $TCN_M^{WP}$ means that only the data of WP are used for training our model. Metrics Distance, Pressure, and Wind are introduced in the section titled 'Metrics'. Best view in color. Source data are provided as a Source Data file.

such as NMPT[31], DLM[27], GBRNN[26], and MMSTN[41]. The third group includes large-scale meteorological models, including Pangu-Weather[32] and Fengwu[33].

LSTM, GRU, and SGAN are classic time-series prediction methods in computer science. All of these methods are open-source, but they only accept one-dimensional data as input. Therefore, we used one-dimensional data as input for these methods, which includes both track and intensity information.

NMPT, DLM, and GBRNN are modifications of time-series networks, such as RNN and LSTM, adapted by previous researchers for TC prediction. These methods are not open-source, but since they use only one-dimensional data and they are relatively simple, we reproduced these models based on their descriptions in the papers and applied them to our dataset. MMSTN is a method among them with open-source code and data (the one-dimensional best-track dataset). It also predicts track, maximum wind speed near the center, and minimum central pressure simultaneously. So we used it as the main comparison method.

Pangu-Weather and Fengwu are designed for global meteorological variable forecasting, such as mean sea level pressure and various variables at different pressure levels. TC track and intensity prediction is not the direct goal of these large models but can be achieved by calculating the meteorological variables they predict. It is worthwhile to compare our model with these large-scale models. The data used by these large-scale models primarily include several variables from ERA5 reanalysis data: geopotential height, specific humidity, temperature, U-wind, and V-wind at pressure levels [50, 100, 150, 200, 250, 300, 400, 500, 600, 700, 850, 925, 1000], as well as surface-level atmospheric data such as 10m U-wind, 10m V-wind, 2m temperature, and mean sea level pressure.

**Quantitative analysis.** As illustrated in Fig. 5.a-1, 2, and 3, we evaluated all models using the mean absolute error for both TC track and

intensity predictions. To ensure a fair comparison, we adopted the experimental protocol established in MMSTN—the prior state-of-the-art approach—and conducted all tests on the WP dataset, consistent with previous TC forecasting studies.

We first examine the performance of four representative models: LSTM[15], GRU[39], NMPT[31], and DLM[27]. These methods either focused on track or intensity prediction in isolation and demonstrated relatively limited performance. In contrast, SGAN[40], GBRNN[26], and MMSTN[41], which integrated both track and intensity data, delivered significantly improved results. This suggests that incorporating diverse feature sources is beneficial for enhancing prediction accuracy.

Our proposed model, $TCN_M$, which integrates GC-Net, Env-Time-Net, and multiple generator modules, outperformed all existing deep learning approaches. For track prediction, $TCN_M$ achieved improvements ranging from 16.4% to 32.7% over MMSTN[41]. Similarly, in intensity prediction, our model delivered a performance gain of 10.6% to 29.1%. Notably, the performance gap widened as the forecast lead time increased.

We are also interested in the performance of large-scale models. The results of Pangu-Weather and Fengwu are shown in Fig. 5.a-1 and a-2. The experimental results show that large-scale models have a significant advantage in short-term (6-24 h) TC track prediction, as shown in sub-figure 5.a-1. However, when predicting TC intensity, both of these large models seem to struggle to provide accurate predictions, as shown in the top part of the sub-figure 5.a-2. The reasons are as follows:

- Reasons for the good track performance of large models: TC tracks are primarily influenced by large-scale atmospheric flows, such as the subtropical high and wind fields. These atmospheric variables are relatively stable and easier to simulate and predict with meteorological models. The Pangu-Weather model effectively captures these large-scale dynamic structures, leading to its strong performance in track prediction.

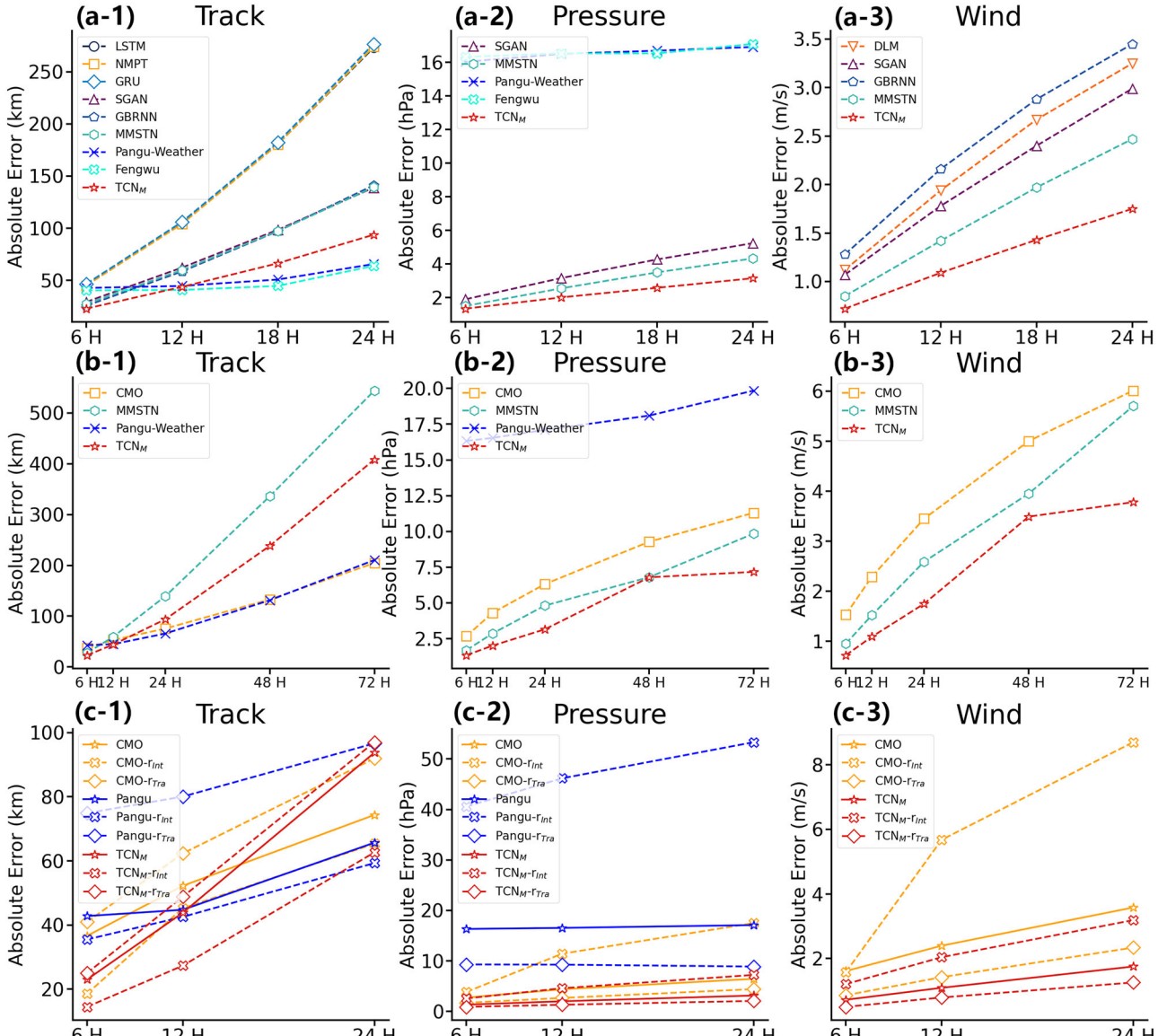

**Fig. 5 | Comparisons of TropiCycloneNet Model (TCN$_M$) with other methods across various aspects.** The comparative results are based on Western North Pacific (WP) sea area data, as most of the comparative methods only provide performance results for the WP region. Sub-figures (**a-1**, **a-2**, **a-3**) present the average absolute error comparisons of tropical cyclone (TC) predictions among different deep learning methods. Sub-figures (**b-1**, **b-2**, **b-3**) show the long-term forecasting (6-72 h) capabilities of TCN$_M$. Sub-figures (**c-1**, **c-2**, **c-3**) demonstrate the performance of TCN$_M$ during rapid changes in TC track or intensity compared to its overall performance on the entire dataset. In these sub-figures, The three solid lines in the represent the results of Chinese Central Meteorological Observatory (CMO), Pangu-Weather, and TCN$_M$ on the complete test set, respectively. CMO-$r_{Int}$, Pangu-$r_{Int}$, and TCN$_M$-$r_{Int}$ represent the results of CMO, Pangu-Weather, and TCN$_M$ on rapid intensifying TC cases, respectively. CMO-$r_{Tra}$, Pangu-$r_{Tra}$, and TCN$_M$-$r_{Tra}$ represent the results of CMO, Pangu-Weather, and TCN$_M$ on curving TC cases, respectively. Best viewed in color. Source data are provided as a Source Data file.

- Reasons for poor intensity performance of large models: Pangu is a global-scale weather prediction model. On a global scale, the evolution of TC intensity is a relatively small-scale process, which may require region-specific models focused on TC prediction for accurate capture the evolution of TC intensity. Therefore, specialized datasets and methods tailored to TC prediction are necessary for better intensity forecasting.
- Reasons for our model's better intensity prediction: Our model not only extracts features from 3D reanalysis data but also learns from direct one-dimensional historical information, such as the TC's past track and intensity. This enables the model to learn how TCs evolve over time. Additionally, our environment module extracts features from historical track and intensity changes within the environmental data of our dataset, allowing the model

to capture more patterns of TC behavior over time. Due to the high computational cost, Pangu uses only current data to predict future values without incorporating historical data. Fengwu uses data from the current and past 6 h (two time steps) to predict future outcomes. In contrast, our model learns from a total of eight time steps, covering 42 h of historical data, allowing it to extract more information.

**Qualitative analysis of TC prediction.** To intuitively demonstrate our method's effectiveness, we visualized track and intensity prediction results and compared the performance of TCN$_M$ with that of the previously best-performing deep learning method, MMSTN[41]. As illustrated in Fig. 6, a sequence of red circles represents the actual track of the TC, while the semitransparent orange area indicates the potential

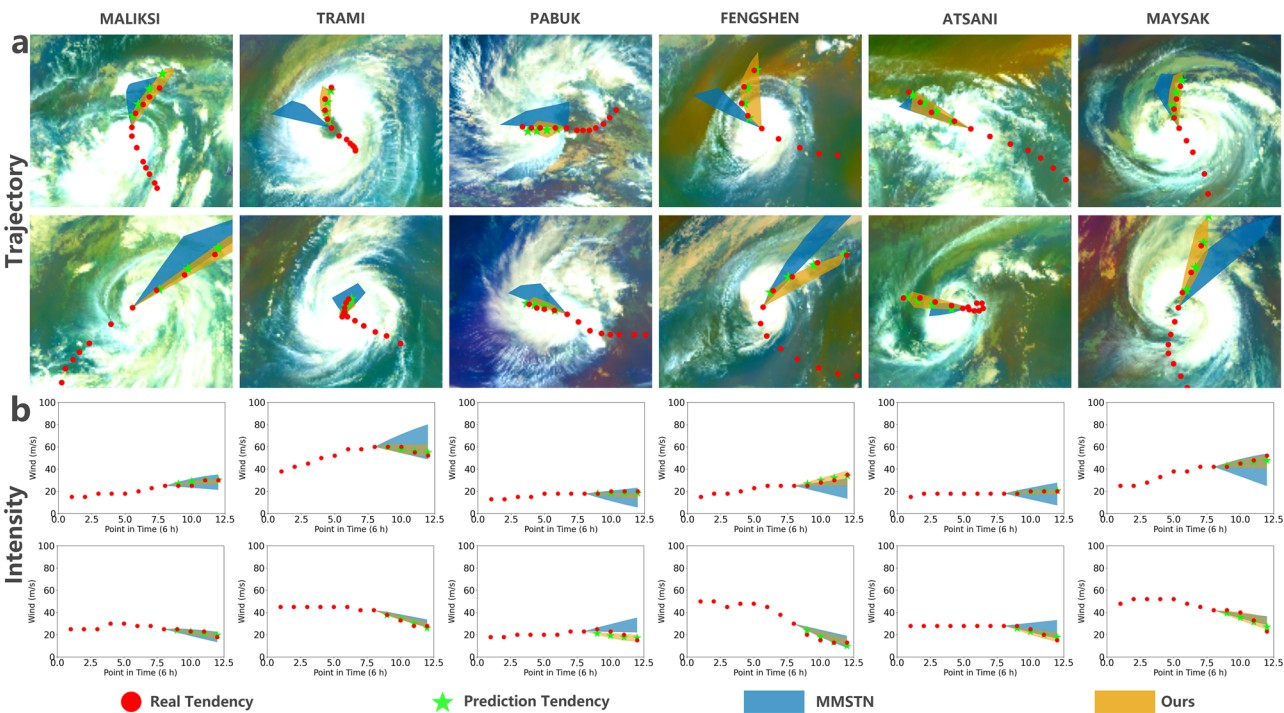

**Fig. 6 | The visualizations of TropiCycloneNet Model's results.** Examples of track and intensity predictions from 6 h to 24 h and the comparison between our method and previous deep learning model (MMSTN) on the potential predictions for six tropical cyclones: MALIKSI (2018, severe tropical storm, Summer), TRAMI (2018, super typhoon, Summer), PABUK (2019, severe tropical storm, Winter), FENGSHEN (2019, strong typhoon, Summer), ATSANI (2020, typhoon, Autumn), and MAYSAK (2020, super typhoon, Summer). Sub-figure **a** shows the track prediction performance of these two models and Sub-figure **b** shows the intensity prediction performance of these two models. Best viewed in color.

trajectories as calculated by our method's multiple predictions, the semitransparent blue region represents the potential tendencies of MMSTN, and a sequence of green stars marks the most accurate predictions from our method, and the prediction results' background displays satellite cloud imagery for each TC.

To demonstrate the superior performance of $TCN_M$, we selected six TCs of varying grades in different years: STS MALIKSI (2018) and PABUK (2019), TY ATSANI (2020), STY FENGSHEN (2019), and SuperTY TRAMI (2018) and MAYSAK (2020). PABUK occurred in winter; ATSANI in autumn; while the other four TCs occurred in summer. The semi-transparent green regions show that $TCN_M$ performs well when it observes the TC impacted by different environmental factors (e.g., the grade and the season of TC). One of the primary motivations for developing the key module Env-T-Net was to enable our model to tackle TCs in varying environments. Otherwise, the difficulty of fore-casting varies among different TCs. For the simpler case of linear prediction, as illustrated by ATSANI's track and intensity image at the first row and the third row respectively, $TCN_M$ achieved results comparable with MMSTN and provided an accurate and small potential region of prediction. In more complex cases of nonlinear prediction, $TCN_M$ offered a broader (when compared with the easy samples) but reasonable prediction. Given that neither meteorological experts nor our method can predict TC futures flawlessly, all we need to do is to provide a reasonable and accurate prediction conditioned on limited information as much as possible. Fortunately, our method, powered by multi-generators and GC-Net, is capable of accomplishing this task. Each generator in our method can extract desired features from his-torical TC data and environmental factors for prediction. This mechanism ensures that each generator's predictions are reasonable and prevents our method from forecasting out-of-distribution (OOD) samples. In challenging cases, each $TCN_M$ generator forms its own prediction tendencies based on known information, creating a more informative and robust forecasting area for meteorologists and

officials. Therefore, the potential prediction area for complex cases is larger than that for simpler ones. Additionally, our method sig-nificantly outperforms MMSTN in complex cases. All these findings further validate the effectiveness of our method.

**Analysis of longer term predictions.** We also aim to explore the longer time forecasting performance of our model. Thus, we obtain the forecasting performance of our model from 6 to 72 h and compare it with one of previous best methods—MMSTN, China Meteorological Observation method CMO, and large weather model Pangu-Weather, which is shown in Fig. 5.b-1, b-2, and b-3. From the results in the table, the proposed $TCN_M$ outperforms MMSTN, CMO, and Pangu-Weather in the task of TC intensity prediction. Our method achieves the best performance in track prediction for the 6-h and 12-h forecasts. How-ever, after 12 h, its performance begins to fall behind CMO and Pangu-Weather, and the gap continues to widen as the prediction time increases.

We believe that the reasons of our model under-performing in long-term tropical cyclone (TC) track prediction are mainly attributed to the accumulated errors in multi-step forecasting. Intensity predic-tion performs better than the track prediction, possibly because track prediction involves forecasting both latitude and longitude, which are two dimensions. As the number of prediction steps increases, the accumulated errors grows exponentially. In contrast, intensity pre-diction involves only a single dimension, making it easier to predict. Although accumulated errors still exist, they grow linearly. Therefore, intensity prediction outperforms track prediction.

**Analysis of rapid intensifying and recurving TC cases.** We also conducted experiments to analyze the performance of our method on rapid intensifying and recurving TC cases, as shown in Fig. 5.c-1, c-2, and c-3. First, we focus on the selection of rapid intensifying cases. Rapid intensification is currently defined as a tropical cyclone (TC)

**Table 4 | Ablation experiments (Top) and comparison with Numerical Weather Prediction models (Bottom)**

| H | M-GC | Env-T | TCN_D | Track (km) | | | | Wind (m/s) | | | |
|---|---|---|---|---|---|---|---|---|---|---|---|
| | | | | 6 h | 12 h | 18 h | 24 h | 6 h | 12 h | 18 h | 24 h |
| | | | | 27.52 | 59.93 | 97.28 | 139.34 | 0.85 | 1.42 | 1.97 | 2.47 |
| ✓ | | | | 25.80 | 50.08 | 75.89 | 105.53 | 0.81 | 1.36 | 1.86 | 2.23 |
| ✓ | ✓ | | | 24.39 | 47.50 | 73.76 | 106.08 | 0.73 | 1.24 | 1.72 | 2.17 |
| ✓ | | ✓ | | 25.02 | 48.50 | 72.89 | 102.28 | 0.85 | 1.36 | 1.77 | 2.07 |
| ✓ | ✓ | ✓ | | 23.84 | 45.03 | 67.93 | 95.39 | 0.72 | 1.15 | 1.48 | 1.81 |
| ✓ | ✓ | ✓ | ✓ | 22.98 | 43.83 | 66.41 | 93.76 | 0.72 | 1.09 | 1.43 | 1.75 |

| Region | Method | Track (km) | | Wind (m/s) | |
|---|---|---|---|---|---|
| | | 12 h | 24 h | 12 h | 24 h |
| NA | OFCL | 39.11 | 58.97 | 2.97 | 4.42 |
| | OCD5 | 77.78 | 172.72 | 3.91 | 6.09 |
| | EMXI | 40.37 | 63.67 | 4.29 | 6.25 |
| | HCCA | 37.26 | 57.23 | 3.27 | 4.44 |
| | TCN_M | 41.79 | 98.22 | 1.35 | 2.14 |
| EP | OFCL | 38.08 | 59.71 | 3.09 | 5.03 |
| | OCD5 | 62.82 | 129.75 | 3.88 | 6.67 |
| | EMXI | 41.19 | 66.34 | 4.59 | 7.35 |
| | HCCA | 36.30 | 56.04 | 3.27 | 4.96 |
| | TCN_M | 29.58 | 70.63 | 1.22 | 1.97 |
| WP | CMO | 52.27 | 74.18 | 2.42 | 3.62 |
| | TCN_M | 43.83 | 93.76 | 1.09 | 1.75 |

[1]Ablation experiments: Receiving Heterogeneous Meteorological Data (H), Multi-Generators with Generator Chooser Network (M-GC), and Environment-Time Network (Env-T). The item of $TCN_D$ means that the results were from our model trained on all the six sea regions data.

[2]Comparison with Numerical Weather Prediction models: Comparison with Official National Hurricane Center forecast (OFCL), Climatology and Persistence model (OCD5), European Centre for Medium-Range Weather Forecasts global model (EMXI), The Hurricane Forecast Improvement Program Corrected Consensus Approach (HCCA), and Center Meteorological Observatory forecast (CMO), on three main sea areas.

experiencing an increase in maximum sustained wind speed near its center by at least 15.432 m/s within 24 h. A typical case is defined as a TC sequence spanning 12 time steps, with a time resolution of 6 h, resulting in a total length of 66 h. Of these, the first 42 h represent the observed TC, while the remaining 24 h are the prediction target. We consider a 48-h time window, which includes 24 h before and after the current time step. If, at any point within this 48-h window, the maximum wind speed near the TC's center increases by at least 15.432 m/s over a 24-h period, we classify the case as a rapid intensifying TC case. In the WP test dataset, there are 2602 cases in total, 192 of which are rapid intensifying TC cases, accounting for 7.37%.

For recurring TC cases, there is no established standard for defining such cases. We set a reasonable criterion to select these cases. Similarly, we use a 48-h time window encompassing 24 h before and after the current time step. Within this window, we calculate the angular change with the TC track every 6 h. If the track experiences at least one angular change of over 70 degrees within any 6-h interval during this window, we classify the case as a recurring TC case. We aim to select ~10% of the samples from test dataset as recurring TC cases according to the criteria we have set. Based on our analysis, there are 279 recurring TC cases, representing 10.72% of all WP cases.

The experimental results reveal an interesting phenomenon. In cases of rapid intensification of tropical cyclones, the performance of CMO shows a more significant decline in predicting intensity (pressure and wind) compared to its performance on the entire test set. In contrast, although our model also experiences a decline, the magnitude of this decline is smaller than that of CMO. The decline of our model can be attributed to the complexity of sudden changes in TC intensity and the smaller proportion of relevant samples in the training set. These factors make it challenging for the model to simulate the rapid intensification process. However, for track prediction, our model demonstrates significant improvement compared to the results from the entire test set. The average rate of change in the track direction in rapid intensifying TC cases is 10.1 degrees per 6 h, and the maximum sustained wind speed near the center is 44.96 m/s. This suggests that rapid intensification often occurs when the TC is already a strong system with a well-developed structure and high intensity. At this stage, the TC's track is less influenced by environmental factors, resulting in nearly linear motion, which is easier to predict, with an average change rate of 10.1 degrees per 6 h. As a result, the model performs well in predicting the track for these cases.

A similar phenomenon is observed in recurring TC cases. The CMO method also shows a significant decline in track prediction performance compared to its performance on the entire test set. In contrast, although our model experiences a decrease in track prediction, the drop is less pronounced than that of CMO and our model demonstrates substantial improvement in intensity prediction compared to its performance across the entire test set. We further analyzed this phenomenon. In recurring TC cases, the average change rate of maximum sustained wind speed near the center is 3.59 m/s per 24 h, and the average TC intensity is 22.21 m/s. The smaller intensity changes in these cases contribute to the model's improved intensity prediction performance. These interesting phenomena could be considered in future work. For instance, rapid intensification is more likely to occur during the high-intensity stage of a TC, whereas track direction changes are more likely during the low-intensity stage.

As shown in Fig. 5.c-1 and c-2, we also show the performance of Pangu-Weather on rapid intensifying and recurring TC Cases. Experimental results demonstrate that under extreme conditions, the performance of large-scale model Pangu-Weather is inferior to ours in both track and intensity forecasting. In particular, Pangu-Weather shows a significant drop in track forecasting accuracy when dealing with recurring TC cases. This indicates that, as a global forecasting model, Pangu-Weather struggles to accurately capture certain extreme

variations in TC systems. In contrast, our model is specifically designed for tropical cyclone prediction and exhibits superior performance under such extreme scenarios. These findings underscore the importance of the method proposed in this paper.

## Ablation studies

We carried out ablation studies to demonstrate the enhancements provided by various modules within TropiCycloneNet to our deep learning approach. Results, presented in Table 4.1, include the impact of the design for receiving heterogeneous meteorological data (H), Multi-Generators with GC-Net (M-GC), Env-Time-Net (Env-T), and the building of dataset $TCN_D$. First, it is clear that integrating H leads to superior performance over the model without it. It means that the design for receiving heterogeneous meteorological data including $Data_{3d}$ and $Data_{1d}$, in $TCN_M$ is effective, resulting in an average improvement of 17.23% in track prediction and 3.35% in intensity prediction. Second, integrating M-GC, known for its advanced learning capabilities, significantly boosts the performance of the model. When M-GC is combined with H, $TCN_M$ achieves an average improvement of 3.22% in track prediction and 11.79% in intensity prediction. Then, we validated Env-T's contribution by comparing methods with H alone to those with both H and Env-T. The Env-T module yielded 3.30% and 2.00% average improvement, respectively. Finally, the $TCN_M$ model trained on $TCN_D$ with the inclusion of H, M-GC, and Env-T, demonstrated significant performance improvements over models lacking these elements.

Furthermore, the creation of $TCN_D$ has provided inspiration for the development of these deep learning modules. For example, when facing the $Data_{3d}$ and $Data_{1d}$ in $TCN_D$, we develop a multi-modal framework to extract enhanced information by incorporating these heterogeneous meteorological data (H). Considering the rich information of the Env-Data, we design the Env-Time-Net to guide our model to make more reasonable predictions by using these environment information. We also believe that $TCN_D$ can lay a solid data foundation for designing even more innovative deep learning models in the future. In summary, the ablation studies confirmed the effectiveness of our design and key modules, as well as the importance of our dataset.

## Comparison with official methods in different sea areas

To better evaluate our model, $TCN_M$, we selected several official forecasting methods for comparison, as shown in Table 4.2. Three of these methods originate from authoritative agencies: the National Hurricane Center, the European Centre for Medium-Range Weather Forecasts, and the CMO. The other two are a statistical method and a consensus method, respectively. Remarkably, our $TCN_M$ model demonstrated superior performance when compared to the Climatology and Persistence model (OCD5), which is a classical statistical method. In fact, $TCN_M$ exhibited the most exceptional performance in TC intensity prediction among all methods evaluated. However, in terms of track forecasting, there remains a gap between $TCN_M$ and the other official methods. Only in 12 h prediction, $TCN_M$ achieved results that were comparable to or even better than those of other methods. Overall, $TCN_M$ is a potential TC prediction method and there are many aspects that are worth exploring in the future.

## Discussion

The building of $TCN_D$ addresses the absence of an open, large-scale, and multi-modal TC prediction dataset. Incorporating diverse meteorological data from six main sea areas worldwide, $TCN_D$ enables deep-learning models to gain a deeper understanding of TC development through exposure to a greater variety and quantity of TCs. Besides, $TCN_D$ eliminates the barrier for AI researchers who lack the time or expertise to construct a high-quality TC dataset, serving as a valuable resource for the AI for TC field. Additionally, $TCN_D$ serves as a benchmark dataset, offering researchers a platform to test innovative

ideas and enhance the application of deep learning in TC forecasting. During building this dataset, we considered the meteorological characterics of TC and select appropriate $Data_{3d}$, $Data_{1d}$, and Env-Data, which means that there is some meteorological knowledge in $TCN_D$. Thus, researchers can draw inspiration for deep learning model design from analyzing $TCN_D$, similar to the design principles behind $TCN_M$. Based on our dataset $TCN_D$, we proposed $TCN_M$, a multi-modal deep learning model of TC prediction. It considers not only the drawbacks of previous deep learning in AI field, like the OOD problem, but also the neglect of meteorology among past deep learning methods, like the environmental factor problem. It is a good baseline model for meteorology researchers to learn how to apply deep learning technologies to TC prediction or other meteorological problems. In summary, TropiCycloneNet bridges the gap between AI and TC forecasting, encouraging interdisciplinary collaboration in this challenging yet rewarding field to collectively advance deep learning methods for TC prediction.

Of course, TropiCycloneNet is not perfect, as there are some drawbacks that need to be improve in the future. As for the $TCN_M$, the long-term track prediction is not satisfied. The primary reasons include (1) a lack of consideration for TC's physical mechanisms in the design of deep learning models, and (2) $TCN_M$'s failure to fully leverage all information in $TCN_D$, despite considering multi-modal data. Exploring how to utilize deep learning technologies to simulate key physical mechanisms of TCs will be a valuable future direction. Other feasible strategy should be mentioned is that we can combine $TCN_M$ with other methods that are good at predicting track of TC, like the Pangu-Weather model and other methods of official meteorological agencies. This strategy could lead to the development of superior models capable of more effectively utilizing $TCN_D$ data for enhanced accuracy in both track and intensity predictions. For the $TCN_D$, although it was developed with the assistance of multidisciplinary experts and meets the requirements of most current deep learning TC methods, there are also some aspects that we have not considered. Feedback from researchers who have used or analyzed $TCN_D$ is crucial for its further improvement.

Furthermore, recent years have witnessed the rapid development of large-scale deep learning models, significantly impacting many aspects of our daily lives, such as weather forecasting. The emergence of large-scale weather models, such as Pangu-Weather[32], Fengwu[33], and Graphcast[34], has led many researchers to adopt an optimistic outlook on the application of deep learning in weather and climate forecasting. These large-scale models have achieved breakthroughs in medium-range global weather forecasting, offering effective pre-trained models for various downstream meteorological tasks, including TC forecasting. TropiCycloneNet, both a benchmark dataset and an innovative method for TC forecasting, will play a crucial role in fine-tuning these large-scale models for TC forecasting, enhancing their performance in this specialized meteorological task.

## Methods
### The framework of TropiCycloneNet model

$TCN_M$ is a GAN-based framework designed for TC forecasting, capable of handling multi-dimensional and multi-modal input data. To efficiently process sequential information from heterogeneous sources, we introduce two specialized encoders: a 3D-Data Encoder for spatiotemporal meteorological data and a 1D-Data Encoder for time series data. To further leverage environmental context, we develop the Environment-Time Network, which captures both instantaneous and evolving environmental conditions over time.

Once comprehensive representations of TC historical behavior and environmental dynamics are extracted, we employ a set of parallel generators $(g_1, g_2, \ldots, g_K)$ to simulate diverse potential evolution paths. The Generator Chooser Network then dynamically selects the most suitable generator by leveraging both environmental and TC-related

features, thereby enhancing the reliability of track and intensity predictions. A discriminator module is incorporated to evaluate the alignment between predicted results and realistic data. To train the model effectively, we apply a combination of loss functions tailored to track and intensity prediction tasks. The overall architecture of $TCN_M$ is depicted in Fig. 2.

**3D-data encoder**. As shown in Fig. 2, the red pathway represents the 3D-Data Encoder[6]. This module performs two primary functions. First, it leverages a 3D-UNet-based architecture[36] to capture rich spatio-temporal representations from past TC sequences. Second, it generates forecasts of future $Data_{3d}$ sequences conditioned on the historical inputs. These predicted sequences are subsequently forwarded to the Decoder-LSTM (De-LSTM) units within the Multiple Generators for downstream track and intensity forecasting.

Why did we design a specific branch to process $Data_{3d}$? There are two reasons. First, CNNs, utilized in this branch, are more suitable than conventional Multi-Layer Perceptrons (MLPs) for extracting features from $Data_{3d}$. This is because $Data_{3d}$ contains rich spatial information, and CNN excels at extracting spatial features through its unique calculations. Additionally, TC prediction is a task that involves temporal prediction using time sequence data. Therefore, 3D-Unet was chosen as the core component of the 3D-Data Encoder to extract crucial spatiotemporal features for accurate TC prediction. The second reason is related to the alignment of inputs between Encoder-LSTM (En-LSTM) and De-LSTM. De-LSTM can obtain the $Data_{1d}$ easily from the output of En-LSTM. But $Data_{3d}$ is difficult to be predicted by the MLP-based En-LSTM. While 3D-Unet is widely used in similar tasks, such as precipitation forecasting[42]. So we build this branch to predict the future $Data_{3d}$ and make these data to be a part of De-LSTM's input. This capability ensures that De-LSTM's input aligns with that of En-LSTM, allowing De-LSTM to access more information for accurate predictions. Overall, the process of the 3D-Data Encoder, including a 3D-Unet (3DUnet) and a Multilayer Perceptron (MLP,$\phi(\cdot)$), can be expressed as follows:

$$\hat{Y}_{3d} = 3DUnet(X_{3d}; W_{3D-Unet}) \tag{5}$$

$$e_{3d}^{En}, e_{3d}^{De} = \phi([X_{3d}, \hat{Y}_{3d}]; W_{MLP_{3d}}) \tag{6}$$

where $X_{3d}$ represents the history sequence $Data_{3d}$ of TC, $\hat{Y}_{3d}$ represents the future predicted $Data_{3d}$ of TC, and $e_{3d}^{En}$ represents the $Data_{3d}$ feature input for the 1D-Data Encoder, $e_{3d}^{De}$ represents the predicted $Data_{3d}$ feature input for Generators. $W_{3D-Unet}$ and $W_{MLP_{3d}}$ represent the weights of the 3D-Unet and MLP respectively. Thus, the feature extraction of $Data_{3d}$ is completed, resulting in features $e_{3d}^{En}$ and $e_{3d}^{De}$.

**1D-data encoder**. As depicted in Fig. 2, the 1D-Data Encoder is represented by the blue branch[6]. This branch is a fundamental component of $TCN_M$. It encodes $Data_{1d}$ and integrates features from the 3D-Data Encoder $e_{3d}^{En}$. First, the 1D-Data Encoder processes $Data_{1d}$ using an MLP to extract features $e_{1d}^{En}$. Then, $e_{3d}^{En}$ and $e_{1d}^{En}$ are concatenated as MLP input to produce $e^{En}$, which is fused with the features from two different dimensions. Finally, En-LSTM extracts temporal features from $e^{En}$. The primary process of 1D-Data Encoder can be expressed as follows:

$$e_{1d}^{En} = \phi(X_{1d}; W_{MLP_{1d}}) \tag{7}$$

$$e^{En} = \phi(cat(e_{1d}^{En}, e_{3d}^{En}); W_{MLP_{fusion}}) \tag{8}$$

$$h_t = \begin{cases} En_{LSTM}(h_0, e_t^{En}; W_{En-LSTM}) & \text{if } t=1, \\ En_{LSTM}(h_{t-1}, e_t^{En}; W_{En-LSTM}) & \text{otherwise}. \end{cases} \tag{9}$$

where $\phi(\cdot)$ is an MLP module and $En_{LSTM}(\cdot)$ represents the function of the En-LSTM. $t \in \{1, 2, \ldots, n\}$ where $n$ is the sequence length of the input data. $e_t^{En}$ denotes the fused features at the time step $t$. $h_0$ denotes an initial status vector initialized to zero and $h_t$ denotes the temporal-spatial features from the first $(t-1)$ time steps of the history $Data_{1d}$ and $Data_{3d}$ of TC. Equation (9) will be executed iteratively for $n$ times to obtain the final temporal-spatial feature $h_n$. $W_{MLP_{1d}}$, $W_{MLP_{fusion}}$, and $W_{En-LSTM}$ are the corresponding weights of the modules.

**Environment-time network (Env-T-Net)**. Unlike $Data_{1d}$ and $Data_{3d}$, the environmental data, as shown in Table 1.3, differs in both content and type, requiring a dedicated encoder for effective feature extraction. To address this, we designed an environmental module called the Environment-Time Network (Env-T-Net). In this module, besides conventional Convolutional Neural Networks (CNN) and Multi-Layer Perceptrons (MLP), we also used a Transformer[38] network for the environmental data. The CNN and MLP are tailored to accommodate environmental data across various modalities and dimensions. The Transformer module captures both temporal relationships and inter-actions between different types of environmental data. This approach enhances the module's effectiveness in extracting critical environmental information. The main process of Env-T-Net is as follows:

$$e_{1d}^{Env} = \phi(X_{1d}^{Env}; W_{MLP-Env}) \tag{10}$$

$$e_{3d}^{Env} = CNN(X_{3d}^{Env}; W_{CNN-Env}) \tag{11}$$

$$e_{Env} = \phi(cat(e_{1d}^{Env}, e_{3d}^{Env}; W_{MLP-Envfusion})) \tag{12}$$

$$e_{Env} - time = T(e_{Env}; W_{Transformer-Env}) \tag{13}$$

where $X_{1d}^{Env}$ and $X_{3d}^{Env}$ are the $Data_{1d}$ and $Data_{3d}$ of environment respectively. $e_{1d}^{Env}$ represents the features of $Data_{1d}$ and $e_{3d}^{Env}$ represents the features of $Data_{3d}$. $W_{MLP-Env}$, $W_{MLP-Envfusion}$, and $W_{CNN-Env}$ are the corresponding weights of modules. $e_{Env}$ represents the features extracted from environment data without time information. $T(\cdot)$ means the Transformer module that used for obtaining the finial features of environment data $e_{Env} - time$. $W_{Transformer-Env}$ is the weights of Transformer module.

**Generator chooser network (GC-Net)**. To mitigate the limitations of single-generator approaches—such as insufficient generalization and vulnerability to out-of-distribution (OOD) inputs—we introduce a multi-generator strategy. Our model dynamically selects an appropriate generator for each prediction instance, guided by both TC-specific and environmental context[6]. Specifically, the probability of selecting generator $g^i$ is modeled as $P^i(g^i, |, h_n, e_{Env} - time)$, where $h_n$ and $e_{Env} - time$ represent the encoded TC history and environmental information, respectively. Since different generators may be more suitable under varying conditions, these selection probabilities differ across predictions. Figure 2 visualizes these probabilities in a roulette-style representation, with $K$ denoting the total number of generators. Each generator captures a distinct forecasting pattern tailored to diverse environmental scenarios. For example, in predicting the track of a TC, if a generator $g^q$ is more adept at predicting a northwest track given a specific environment $env^i$, and our method also considers the future direction of the TC track as northwest in the environment $env^i$, the probability $P(g^q|h_n, e_{Env} - time)$ should be higher for $g^q$ than for the other generators. This implies that $g^q$ has the highest selection probability among the generators. In such cases, our predictions become more precise and reasonable. To calculate each generator's probability, we designed the GC-Net. GC-Net takes multi-modal features $h_n$ and $e_{Env-time}$ as inputs and outputs the probabilities for each

generator. The main process of GC-Net is:

$$P = \phi(\text{cat}(h_n, e_{\text{Env}} - \text{time}; W_{\text{MLP-GCNet}})) \tag{14}$$

Here, $P$ represents the probability array for each generator, where $\phi(\cdot)$ is an MLP module, and $W_{\text{MLP-GCNet}}$ denotes the weights within the GC-Net. Subsequently, we employ the Monte Carlo method[43] to determine the identification list $gid_{list}$ of the selected generators based on $P$ and the sampling number $l$. The length of $gid_{list}$ equals $l$.

**Multiple generators.** Although each generator shares an identical network architecture, they are independently parameterized to capture diverse prediction behaviors under varying conditions[6]. This diversity is visually illustrated in the left subplots of Fig. 2, where each generator demonstrates a distinct forecasting tendency. The core of each generator is a Decoder-LSTM (De-LSTM) module, which transforms encoded features into future TC states. During inference, generator selection is guided by a predefined generator identity list, denoted as $gid_{list}$. The selected generator receives multi-source feature inputs: $h_n$ from the 1D-Data Encoder, $e_{3d}^{\text{De}}$ from the 3D-Data Encoder, and $e_{\text{Env}} - \text{time}$ from the Environment-Time Network. To model prediction uncertainty, the generator also incorporates a latent noise vector $z \sim \mathcal{N}(0, 1)$[40].

First, model fuses the features $h_n$ and $e_{\text{Env}} - \text{time}$ by using an MLP ($\Phi(\cdot)$) to obtain fused features $h_{\text{fused}}$, as shown in Equation (15). Then, $h_{\text{fused}}$ and the noise vector $z$ are concatenated and used as the initial status of Decoder. The model then selects a generator ($g^i$) by GC-Net and Monte Carlo method. $[h_{\text{fused}}, z]$ and the latest fused history data $e_n^{\text{En}}$ are input into decoder function $De_{\text{LSTM}^i}(\cdot)$ of $g^i$ to predict track and intensity of the TC $\hat{Y}_{n+1}$, as shown in Equation (16) (t = 1). We only predict 1D data, such as the track and intensity of TC, and do not predict 3D data. We believe that 3D data are more challenging to predict than 1D data, and the main task of our work is to predict the track and intensity of the TC, which are 1D data. Predicting 1D and 3D data simultaneously affects the performance of 1D data prediction. Therefore, we only predict the track and intensity of TC. However, if we want to predict the track and intensity after the $n + 1$ time step, we need to encode $\hat{Y}_{n+1}$ using an MLP to obtain 1D features and fuse them with the 3D predicted features $e_{3d}^{\text{De}}$ from Equation (6), as shown in Equations (17) and (18). We align the input types of En-LSTM and De-LSTM with both 1D and 3D features by the approach mentioned above. This method reduces the complexity of our model's prediction task by focusing primarily on 1D data prediction, while also providing the generator with additional information for more accurate predictions. The main process of predicting is:

$$h_{\text{fused}} = \phi(h_n, e_{\text{Env}} - \text{time}; W_{h_f}) \tag{15}$$

$$\hat{Y}_{n+t}^i, h_{n+t}^i = \begin{cases} De_{\text{LSTM}^i}([h_{\text{fused}}, z^i], e_n^{\text{En}}; W_{\text{LSTM}_{\text{De}}^i}) & \text{if t} = 1, \\ De_{\text{LSTM}^i}(h_{n+t-1}^i, e_{n+t-1}^{\text{De}}; W_{\text{LSTM}_{\text{De}}^i}) & \text{otherwise}. \end{cases} \tag{16}$$

$$e_{n+t}^{\text{De}_{1d}} = \phi\left(\hat{Y}_{n+t}^i; W_{e_{1d}^{\text{De}}}\right) \tag{17}$$

$$e_{n+t}^{\text{De}} = \phi\left(\text{cat}(e_{n+t}^{\text{De}_{1d}}, e_{n+t}^{\text{De}_{3d}}; W_{e^{\text{De}}}\right) \tag{18}$$

where $i$ indicates that $g^i$ is selected. $h_{n+t}^i$ denotes the temporal-spatial features from the first $(n + t - 1)$ time steps of $Data_{1d}$ and $Data_{3d}$ of TC, $t \in \{1, 2, \ldots, m\}$. $m$ is the number of time steps for TC track and intensity that we want to predict. $e_{n+t}^{\text{De}_{1d}}$ and $e_{n+t}^{\text{De}_{3d}}$ are the encoded features of predicted 1D and 3D data at future time step $t$ respectively. $e_{n+t}^{\text{De}}$

represents the fused feature of $e_{n+t}^{\text{De}_{1d}}$ and $e_{n+t}^{\text{De}_{3d}}$, which is input into $De_{\text{LSTM}^i}(\cdot)$ to further predict the track and intensity of TC, as shown in Equation (16) (others). Equations (16), (17), and (18) are executed sequentially for $m$ iterations to obtain the TC track and intensity prediction $\hat{Y}_i = \{\hat{Y}_{n+1}^i, \hat{Y}_{n+2}^i, \ldots, \hat{Y}_{n+m}^i\}$. $W_{h_f}$, $W_{\text{LSTM}_{\text{De}}^i}$, $W_{e_{1d}^{\text{De}}}$ and $W_{e^{\text{De}}}$ are the corresponding weights of modules. We have now completed generating the future tendencies of the TC, concluding the generator phase. Next, we proceed to describe the discriminator phase.

**Discriminator.** To enhance the realism of the generators' predictions, we introduce a Discriminator designed to analyze its input features and determine whether the input data correspond to actual TC data: $\text{TC}_{\text{real}} = [X, Y]$, or predicted TC data: $\text{TC}_{\text{fake}} = [X, \hat{Y}]$. The goal is for the Discriminator to discern the rules of TC development during training, identifying any implausible tendencies as 'Fake'.

## Loss functions

Following the detailed overview of the key modules in our method, we now understand the architecture of $\text{TCN}_M$ and the rationale behind its design. To optimize our model, we employ four loss functions: adversarial loss, best of many loss, 3D-Data loss, and GC-Net loss.

**Adversarial loss.** Adversarial loss is a cornerstone of GAN-based models. In our approach, we utilize the original adversarial loss formula to refine both the Generators G and the Discriminator D. Our goal is for the output $\text{TC}_{\text{fake}}$ produced by G to be indistinguishable by D from $\text{TC}_{\text{real}}$, encouraging D to accurately identify any implausible $\text{TC}_{\text{fake}}$ among all inputs it receives, and thereby guiding G to create more realistic and precise predictions of future TC trajectories. The formula for adversarial loss in our method is as follows:

$$\mathcal{L}_{\text{adv}} = \text{CE}(D([X, Y]), T) + \text{CE}(D([X, G(X)]), F) \tag{19}$$

Here, CE denotes the cross-entropy loss function, with $D(\cdot)$ producing the probability that its input is true. $G(\cdot)$ generates predictions for future TC. The pair $[X, Y]$ signifies $\text{TC}_{\text{real}}$, whereas $[X, G(X)]$ denotes $\text{TC}_{\text{fake}}$. Additionally, $T$ represents the label for $\text{TC}_{\text{real}}$, and $F$ denotes the label for $\text{TC}_{\text{fake}}$.

**Best of many loss.** The best of many loss is commonly employed in pedestrian track prediction tasks[40]. In our approach, we select the most accurate prediction from the outputs of the generators and compute the loss relative to the ground truth. This method motivates $\text{TCN}_M$ to generate a range of plausible future TC tendencies. The formula for this loss is defined as follows:

$$\mathcal{L}_{\text{BMS}} = \min_l \left\| Y - \hat{Y}_i \right\|_2 \tag{20}$$

where $i \in gid_{list}$, $l$ is the number of samples, and $\|.\|_2$ is the $l_2$-norm function.

**3D-data loss.** To enhance the extraction of spatiotemporal features from $Data_{3d}$, we incorporate 3D-Data loss to further refine the 3D-Data Encoder. As previously mentioned, we employ a 3D U-Net to predict future $Data_{3d}$ sequences, denoted as $\hat{Y}_{3d}$. The loss is calculated pixel-wise between $\hat{Y}_{3d}$ and the actual $Data_{3d}$ sequence $Y_{3d}$, with the main process outlined as follows:

$$\mathcal{L}_{3D} = \frac{1}{hw \times c \times m} \left\| Y_{3d}^{hw \times c \times m} - \hat{Y}_{3d}^{hw \times c \times m} \right\|_1 \tag{21}$$

Here, $hw$ represents the spatial dimensions of $Data_{3d}$, $c$ indicates the number of channels, $m$ denotes the length of the prediction sequence, and $|.|_1$ refers to the $l_1$-norm function. Overall, the training objective for

the GAN component of $TCN_M$ is summarized as follows:

$$\min_G \max_D \mathscr{L}_{adv} + \lambda_{BMS} \mathscr{L}_{BMS} + \lambda_{3D} \mathscr{L}_{3D} \qquad (22)$$

**GC-net loss.** It is crucial for GC-Net to select appropriate generators based on the specific environmental conditions and TC characteristics. To optimize the GC-Net module, we employ GC-Net loss. This involves using the actual future TC data $Y$ and the predicted TC data $\hat{Y}$ to approximate the likelihood of a specific generator's distribution $P_i$. The primary process is as follows:

$$P(Y|h_n, e_{env-time}, g) \propto \frac{1}{l} \sum_{i \in gid_{list}} \exp\left(-\frac{\left\|\hat{Y}_i - Y\right\|_2^2}{2\sigma}\right) \qquad (23)$$

Subsequently, we calculate the conditional probability of each generator using Bayes' Rule, as follows:

$$P(g|h_n, e_{env-time}, Y) = \frac{P(Y|h_n, e_{env-time}, g)}{\sum_{i=1}^K P(Y|h_n, e_{env-time}, g_i)} \qquad (24)$$

In the final step, we optimize GC-Net by employing cross-entropy loss between the distribution $P(g|h_n, e_{env-time}, Y)$ and the GC-Net's output to optimize GC-Net:

$$\mathscr{L}_{GC} = H(P(g|h_n, e_{env-time}, Y), GC(h_n, e_{env-time})) \qquad (25)$$

where $GC(\cdot)$ is the probability array of each generator by GC-Net.

**Training scheme.** The effectiveness of the generators in making predictions is significantly influenced by the output $P$ from GC-Net. The more accurately our model selects suitable generators, the more precise the predictions it produces. Therefore, we initially train GC-Net using Equation (25), while keeping the rest of our model's parameters fixed for the first $q$ epochs, where $q$ is a predetermined hyperparameter. This initial step allows GC-Net to establish a generator selection strategy with initialized parameters, facilitating the subsequent training of our model's GAN component. Following this, we proceed to train all parameters of $TCN_M$ and achieve optimal performance by employing both Equations (25) and (22).

We deployed $TCN_M$ on the PyTorch platform and executed it on an NVIDIA RTX A6000 GPU. For model optimization, we utilized the Adam optimizer[44] with an initial learning rate of 0.0001. $TCN_M$ was trained using a batch size of 96 over $q+100$ epochs, with the hyperparameter $q$ set to 2. We configured the model with 6 generators ($K = 6$) and a sampling number ($l$) of 6. In our experiments, we fed the model data from the past 48 h ($n = 8$) to predict the TC's future 24-h ($m = 4$) track and intensity. Further details on the experimental setup are available in the code.

## Data availability
TropiCycloneNet has been released on Zenodo and Github, including the dataset $TCN_D$ and the code of $TCN_M$. The code have been opened on Github: https://github.com/xiaochengfuhuo/TropiCycloneNet and Zenodo: https://doi.org/10.5281/zenodo.15024028. The $TCN_D$ is also released on Github: https://github.com/xiaochengfuhuo/TropiCycloneNet-Dataset and Zenodo: https://doi.org/10.5281/zenodo.15009527. The data in TropiCycloneNet are all collected from the open data source. The original data of $Data_{1d}$ are collected from open best track datasets – The China Meteorological Administration Tropical Cyclone Best Track Dataset (CMA-BST, https://tcdata.typhoon.org.cn/en/zjljsjj.html) and International Best Track Archive for Climate Stewardship (IBTrACS, https://www.ncei.noaa.gov/products/international-best-track-archive). $Data_{3d}$ are from the fifth-generation atmospheric reanalysis of the global climate (ERA5,

https://cds.climate.copernicus.eu/datasets/reanalysis-era5-pressure-levels?tab=overview)[35] by the European Centre for Medium-Range Weather Forecasts (ECMWF). Env-Data were analyzed and calculated by our research group and will released in the $TCN_D$. Besides, we also upload a documentary to introduce the detail of $TCN_D$ and how to use this dataset on Github and Zenodo. Source data are provided with this paper.

## Code availability
The $TCN_M$ is publicly available at Github repository (https://github.com/xiaochengfuhuo/TropiCycloneNet) and Zenodo (https://doi.org/10.5281/zenodo.15024028).

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

## Acknowledgements

This work is partially supported by Zhejiang Provincial Natural Science Foundation of China under Grant (No. LRG25F020002, LR21F020002, C.B.), Distinguished Young Scholar of Shandong Province under Grant (ZR2023JQ025, J.L.Z.) and National Natural Science Foundation of China under Grant (No. 62202429, P.M.), (No. U24A20221, P.M. and J.L.Z.), (No. 62020106004, S.Y.C.).

## Author contributions

C.H. and P.M. wrote the manuscript. P.M., C.B., J.L.Z., and S.Y.C. conceived the project framework and defined the research scope. C.H. designed the multimodal deep learning architecture and conducted experiments. C.H., P.M., S.X.C., S.Q.Z., and H.T.Y. were responsible for data collection and preprocessing. C.H., P.M., and J.L.Z. analyzed the results of the proposed model. C.B. proposed the idea of this project and supervised all aspects of this project. All authors reviewed and commented on the manuscript.

## Competing interests

The authors declare no competing interests.
