## [Transparent Peer Review file · Nature Communications]

Benchmark Dataset and Deep Learning Method for Global Tropical Cyclone Forecasting

Corresponding Author: Professor Cong Bai

Version 0:

Reviewer comments:

Reviewer #1

(Remarks to the Author)

The work presented in the manuscript is interesting as well as necessary for the improvement of tropical cyclone (TC) forecasting accuracy, as deep learning methods have inherent capabilities of solving problems where variables are interconnected but in a non-linear way.

Comments on key results:

The proposed deep learning method (TropiCycloneNet) is capable of producing TC track and intensity forecasts at the same time, which has made this model practical from an operational viewpoint. Prediction performance of the method is better while compared with previously well-known deep learning methods in the Western North Pacific. The proposed model also produced better track and intensity forecasts compared to most of the mentioned operational techniques (Table 3) in the North Atlantic, Western North Pacific, and Eastern North Pacific basins for the next 12- and 24-hours. These capabilities obviously support the proposed method to become a potential candidate for operational TC track and intensity forecast. However, the authors could train and test their model for longer time periods, which is considered essential for better emergency management. Without TC track and intensity forecasting results for longer time periods, it seems difficult to comprehend the method's performance for >24h. As a TC spends most of its life span over ocean and sea waters, this capability seems important. Warning residents living in the high TC risk zones before 24h may not be sufficient to save their lives and reduce property damage, which is the ultimate goal of all TC forecasting and warning systems.

Comments on validity:

In lines 313 and 314, the authors mentioned, environmental data was extracted from the 1D and 2D datasets. As the 1D and 2D datasets already contain the environmental properties used in this work, why were those properties extracted and then used again in the prediction process? More clarification would help the readers to understand this part in a better way. Whether the variables were considered for a single level or from multiple vertically arranged pressure levels is not mentioned. Though the authors mentioned about 500 hPa geopotential height data, has it been used in the work is not clear. The authors could provide a complete list of variables under each dataset (can be in a table) to overcome this misunderstanding, which may arise among the readers. Within a TC, if variables from a single pressure level are used, then it is better to be from the steering level. Currently, from which level meteorological and environmental variables are considered seem lacking.

Comments on significance of the work:

The authors used three different sub-modules within their framework. These frameworks are well-known to process information and produce good results individually. Accumulating freely available data under a multi-modal dataset and combining well-established modules under a framework for processing those data seems to be the primary novelty of this work. Predictors the authors mentioned are already in use by various operational TC/weather forecasting models (either individually or combinedly). However, using a deep learning method for processing this dataset is obviously a considerable achievement, as these methods are skillful in processing non-linear relationships (a common phenomenon in a TC system).

Comments on data and methodology:

I would like to appreciate the authors' efforts to address the meteorological context of a TC in their work. However, they

could improve it through linking the datasets with TC properties in methodology. In particular, the authors could mention which type of data is serving which purpose in the developed framework. How 1D and 2D datasets contributed to TC track and intensity prediction appears to be an important thing for the readers to understand.

Description of training and test datasets in lines 354-357 could be clarified through adding better description of the datasets.

>> Whether the datasets were arranged sequentially following the TC formation time and lifespan.

>> Whether TC lifespans were divided to prepare train and test datasets.

>> Why was data collected between 1950 and 2016 used for both training and testing?

>> Why was data collected during 2017 and 2021 used only for testing?

Suggestions for further improvements:

>> Context of using the datasets (role of individual variables in the forecasting process) needs to be improved.

>> As mentioned earlier, from which pressure level/levels the predictors were considered needs to be added.

>> 1D and 2D datasets are already containing environmental information of a TC. Why the authors are extracting this needs to be clarified.

>> What was the performance of the method in case of rapid intensifying and recurring TCs? The authors may add this information, as these TCs pose extra complexity in forecasting end and threat in emergency management end.

Comments on clarity and context:

The authors may follow the clarity and context issues mentioned in the previous paragraphs to make the manuscript more interesting and easily understandable for the readers.

Comments on reference use:

The authors could cite a few more works related to TC meteorology.

(Remarks on code availability)

Reviewer #2

(Remarks to the Author)

This paper presents a new database and deep learning architecture for tropical cyclone trajectory and intensity estimation. It incorporates 2D+t fields from reanalysis, as well as 0d+t data (passed trajectory, intensity, etc.), from worldwide regions. The novelty resides in the release of the database as well as in the new framework, made by different sub-networks involving convolutional neural networks, LSTMs, MLPs and a roulette generators module. This enables to produce a variety of trajectories (i.e. a probability cone) and not only a single trajectory, which is often more useful for forecasters.

The paper is interesting, the release of the database also. I am not fully convinced that the paper is suited for Nature Communications as the added values are after all (rather) incremental. Indeed, the database is useful but 1) it is rather similar to previous studies (which might not have released it), so it is just a matter of re-extracting reanalysis using IBtracs catalog: useful, but not novel 2) the formatting of the data is very important for a database to become a proper benchmark while it is not described, and the many important information are not explained (the size of the 2D data, the list of the proper fields, etc.) and the data is not released yet. As of the architecture proposed, while it seems interesting, it is pretty complex for the results obtained: it does not seem to compete at all with physical current forecasting models, and are not compared with other state-of-the-art models than their previous paper, nor to recent weather forecasting large models (e.g. GraphCast) able to also predict TC forecasting with good accuracy (as mentioned in the state-of-the-art section of the paper). The description of the architecture lacks clarity, and the different sub-networks are not well explained (in particular their differences). To sum up the study is interesting, and worth publishing, but I am not sure it would raise up to the level of Nature Communications as it seems incremental. I would recommend submitting to another journal. In case the editor would rather select 'major revisions', here are my more detailed comments.

- What you call 'Environmental' input information is not well defined, and can be very confusing. In particular, the sentence 'Previous deep learning methods often overlook the impact of environment-related factors, which are critical in the development of TC' is not clear, what do you mean by that? Also, it is not clear the link between env. data and the 1d / 2d data used in the other branches of your model. Please explain.

- When talking about the 'Data 1d', it is slightly misleading (not in 1 dimension but in 0, no?) - or it is 1d if considered time series (but in that case the others are 2D+t)?

- Meteorological data : please detail, I guess you are meaning that you are using reanalysis, but give details about the database (ERA5?). Are these data available in near real-time? Could your model be thus used in an operational way?

- Please clarify which features, how many time steps, what is the size of the window, if it is centered on the TC (if so it means the window moves with time)? Do you think that a fixed window would be better?

- Why did you use old data, i.e. prior to 2000, since the reanalysis and also the 1d data is most probably of bad quality?

- why don't you use any data after 2021?

- how many time steps are used in the passed? [this information was found at the end of the methods section, but it would be

important to have it in the main text and earlier]

- state of the art: MMSTN is not part of your state-of-the-art section while you are using to compare later on. Please add it and explain what it is. Also, by looking at the original MMSTN paper, it does not seem to be the best in TC tracking. So I would suggest comparing with other models. More generally, it is not clear why you chose to compare against these architectures and not others? What do each of them use as input? Did you compare with an architecture using 2D data? Did you compare with recent meteorological forecasting methods like GraphCast that you describe in your introduction?

- Have you considered using a self-supervised task in order to improve the extraction of reanalysis features (since the TC data time steps is rather small)?

methods section:

- data2D branch: the 2 steps and the 2 arrows (in the schema, the En-LSTM and the red embedding) are not clear. Explain how is made the 'embedding' after the Unet, and what is passed in the En-LSYTM branch. Why do you need to input it also in the blue embedding network, and how do you input 2D features with the trajectory/intensity data? Why isn't the red branch enough to take 2d data into account? How do you 'ensure that the De-LSTM input aligns with En-LSTM'? For me, the De-LSTM inputs is composed of En-LSTM outputs, and 2D-embedding outputs, no?

- the 'Encoder 2d' in eq. 1 is not visible in the schema. Maybe consider adding sub-schemas to complement the methods section?

- line 702. 'Then, e2d and e1d are concatenated as MLP input to produce efusion, which is fused with the features from two different dimensions'  what 2 dimensions are you talking about?

- Environmental module. The text lacks of clarity, in particular in order to understand the difference between this and the other networks, especially because you are using the data1d and data2d also..?

- please indicate the size of the 2D windows, is it centered, not centered, and on what?

- 'monte-carlo roulette generator': any citations?

- are the different generators different? why? how does the discriminator help, could you not just perform a supervision on the values?

- 'TCNM was trained using a batch size of 96 over 100+q epochs, with the hyperparameter q set to 2'  ? why using a parameter q if it is just 102 in the end?

- Please add a part about the format of the dataset, its functionality, etc. It is not explained and it not able to look at it for now as it is not released yet

Typos: It is ok to use ChatGPT or another LMM to help you (I've put random sentences of the paper in the GPTZero checking website, it gave me 98% chances of being completely written by an LMM..), but please do proofread afterwards:

- I200 The TCND encompasses includes

- I205 The TCND comprises not only includes the inherent attribute data of TC

- I261 According Fig. 3  to

- Some of there factors

- I426 First, Look at these four methods

- I524 significantly boosts performance of model

- I678 While 3D-Unet is wildly used in similar tasks, such as precipitation forecasting (Agrawal et al. 2019).

- 850 Overall, the overall training objective

- WEncoder2d denotes the weight of 3D-Unet and MLPs  weights

- I742 finial

(Remarks on code availability)

Version 1:

Reviewer comments:

Reviewer #1

(Remarks to the Author)

The authors have revised the manuscript according to the previously provided comments.

However, as this method use input information from TC centered 20° x 20° domain, predicting track with improved accuracy will always be restricted. As large-scale atmospheric flows influencing TC track are relatively stable, the authors could think

of parameterizing it into their model domain for improving track performance.

(Remarks on code availability)

The README file provide necessary instructions for installing and running the application. I configured and installed the code but did not run it using datasets provided in google drive.

Reviewer #2

(Remarks to the Author)

First, I would like to thank the authors for improving the clarity and for adding some additional comparisons, it clearly improved the paper. With this new information, I would like to keep my first advice, i.e. the work is worth publishing without much more editing, but it is not at the level of Nature Communications. I would suggest for example Communications Earth&Environment.

My suggestion is based on:

1) the fact that the trajectory forecast results of new Weather forecast models are substantially better than this paper's results (new figure 5). The long-term forecast figure (5b) does not contain them, but I guess the deep learning weather forecast would also surpass the other methods. The intensity forecast is thus the added value, but it makes the paper weaker.
2) The database, now clearer and accessible, is still not a proper benchmark because the dataset is stored on a google drive and not on a repository with a DOI and thus permanent link (such as Zenodo), and the authors want to keep updating it. I see the value of it, of course, but it means that it cannot be used for a long-term benchmark as it could be removed/modified. As the database is made of open-source fields, not novel by themselves except the environmental fields, the added value of the repository is only if it can become a new solid benchmark dataset.

(Remarks on code availability)

I just checked quickly the repository, it looks correct with some explanation on how to use it. Yet all comments are for now in chinese only. The authors said that they would clean the code afterwards

Version 2:

Reviewer comments:

Reviewer #3

(Remarks to the Author)

From a third-party perspective, the review process for the TropiCycloneNet project reveals a constructive dialogue concerning the contributions and limitations of both the proposed benchmark dataset and the deep learning forecasting method. Reviewers raised pertinent questions that pushed the authors to clarify and strengthen key aspects of their work, particularly regarding data novelty, usability, and the specific merits of their forecasting approach.

1) Regarding the added value of the intensity forecast, Reviewer #2 questioned its significance given the superior track forecasting of newer large-scale models. The authors countered this by emphasizing the substantial difference in computational resources required for their model compared to models like Pangu-Weather, highlighting their approach's accessibility. Crucially, they presented evidence that while large models may excel in overall track prediction, they struggle with TC intensity forecasting, particularly in challenging scenarios such as rapid intensification and recurving TCs. The authors demonstrated their model's superior performance in these critical intensity and extreme track change events, thereby underscoring the contribution of their specialized method.

2) About the Usefulness of the Presented Benchmark Dataset, Reviewer #2 acknowledged the usefulness of the dataset but questioned its novelty, stating it seemed similar to previous unreleased datasets. The dataset's novelty by highlighting its broader temporal coverage (nearly 70 years), wider range of ocean basins (six main sea areas), and richer, more diverse data, specifically the inclusion of environmental data, which is rarely incorporated in previous deep learning approaches. While the constituent data sources (like IBTrACS and ERA5) are widely used, the novelty lies in the idea of compiling and releasing such a large-scale, multi-modal dataset.

3) Based on the review exchange, it appears the authors have addressed the fundamental issues raised by Reviewer #2, including the clarity and accessibility of the dataset and the specifics of the model architecture and inputs. While Reviewer #2 initially felt the dataset was not a "proper benchmark" due to its storage location and potential for modification, the authors' action of uploading it to Zenodo with a DOI addresses this concern about permanence and usability as a benchmark.

4) While the work is strong, a few aspects are worth noting: (1)Long-Term Track Prediction Limitation: TCN_M's track prediction accuracy diminishes for longer lead times (beyond 12 hours) compared to some official and large-scale models. The authors acknowledge this and suggest it as an area for future work, possibly by incorporating more physics-based constraints or hybridizing with other models.(2)Input Domain for Track Forecasting: As pointed out by Reviewer #1, the TC-centered $20^{\circ} \times 20^{\circ}$ input domain for 3D data might inherently limit track prediction accuracy by not capturing broader atmospheric steering flows. The authors defend this by citing computational efficiency, focus on core TC features, and the

Env-T-Net's role in supplementing larger-scale environmental information.

In conclusion, the review process, appears to have significantly improved the manuscript's clarity and the project's accessibility. While initial questions about novelty and comparative performance were raised, the authors provided compelling arguments and took concrete steps (like the public dataset release on Zenodo) to substantiate their claims of contribution. The project offers a valuable open multi-modal dataset and a deep learning model that appears particularly strong in TC intensity forecasting and performance under extreme conditions.

(Remarks on code availability)

Dear Reviewers:

We would like to express our sincere gratitude to the reviewers for recognizing the value of our work, as well as for pointing out the shortcomings in our manuscript. These suggestions will greatly improve the quality of our manuscript, and the revised contents will be highlighted in red font in the manuscript. In this response letter, we would like to address the common issues that were raised by both reviewers first (from pages 1 to 6 in this response letter). Then, we will respond to each of the comments from Reviewer #1 (from pages 6 to 10) and Reviewer #2 (from pages 10 to 25) individually.

For Common Comments:

Both reviewers have mainly identified two important issues: **comparison with more state-of-the-art meteorological models** and **giving more detail of the proposed dataset**.

The first issue is that we did not compare our model with the current state-of-the-art meteorological models, such as GraphCast. Since GraphCast has not fully open-sourced its TC forecasting code, we are currently unable to perform a direct comparison. However, based on the WeatherBench 2, it has been shown that Pangu-Weather have forecasting capabilities comparable to GraphCast. Thus, in this revised version, we have included the results of these two meteorological models, Pangu-Weather and Fengwu (a similar large-scale weather model with Pangu-Weather), for the years covered by our test set. The experimental results are shown in Figure 5.a-1, a-2, and a-3 on page 10.

The results of Pangu-weather and FengWu are shown in the red boxes. The experimental results indicate that our model outperforms the large-scale models in track prediction for the 6-12 hour range and in forecasting the minimum central pressure for intensity prediction (these large-scale models do not predict the wind of TC). While the large-scale models demonstrate strong overall performance in track prediction, they seem to struggle with intensity prediction, where they fail to work properly. For the wind prediction, these two large-scale model do not provide the results of wind. We have provided an analysis of this phenomenon on pages 9-10, lines 544-584, and the description is presented as follows:

We are also interested in the performance of large-scale models. The results of Pangu-Weather and Fengwu are shown in Fig.5.a-1 and a-2. The experimental results show that large-scale models have a significant advantage in TC track prediction, as shown in sub-figure 5.a-1. However, when

predicting TC intensity, both of these large models seem to struggle to provide accurate predictions, as shown in the top part of the sub-figure 5.a-2. The reasons are as follows:

- **Reasons for the good track performance of large models:** TC tracks are primarily influenced by large-scale atmospheric flows, such as the subtropical high and wind fields. These atmospheric variables are relatively stable and easier to simulate and predict with meteorological models. The Pangu-Weather model effectively captures these large-scale dynamic structures, leading to its strong performance in track prediction.
- **Reasons for the poor intensity performance of large models:** Pangu is a global-scale weather prediction model. On a global scale, the evolution of TC intensity is a relatively small-scale process, which may require region-specific models focused on TC prediction for accurate capture the evolution of TC intensity. Therefore, specialized datasets and methods tailored to TC prediction are necessary for better intensity forecasting.
- **Reasons for our model's better intensity prediction:** Our model not only extracts features from 3D reanalysis data but also learns from direct one-dimensional historical information, such as the TC's past track and intensity. This enables the model to learn how TCs evolve over time. Additionally, our environment module extracts features from historical track and intensity changes within the environmental data of our dataset, allowing the model to capture more patterns of TC behavior over time. Due to the high computational cost, Pangu uses only current data to predict future values without incorporating historical data. Fengwu uses data from the current and past six hours (two time steps) to predict future outcomes. In contrast, our model learns from a total of eight time steps, covering 42 hours of historical data, allowing it to extract more information.

The second important issue is the lack of detailed description of the dataset in the manuscript, which makes readers feel confused. In this revised version, we have created a table and added additional descriptions to provide a detailed explanation of what our dataset includes and the role of each variable in TC forecasting. Additionally, we have released the TCN_D dataset along with the corresponding usage instructions on GitHub. To facilitate the review process, we provide both the full dataset and the separate test dataset for download. For more details, please visit: <https://github.com/xiaochengfuhuo/TropiCycloneNet-Dataset>. A more detailed introduction will be updated on GitHub once this work is accepted. The table related to the dataset in the manuscript Table 2 on page 7. We also show the table as follows:

Data Name	Sea surface temperature	Geopotential height	U-component of wind	V-component of wind
Spatial resolution	0.25°	0.25°	0.25°	0.25°
Time resolution	6 hourly	6 hourly	6 hourly	6 hourly
Pressure Level (hPa)	/	200, 500, 850, 925	200, 500, 850, 925	200, 500, 850, 925
Size (around the TC center)	81*81 (20°*20°)	81*81 (20°*20°)	81*81 (20°*20°)	81*81 (20°*20°)

ID	LONG	LAT	PRES	WND	YYYYMMDDHH	Name	Key	Value
6	-10.42	2.68	0.8	-1	2019090112	LINGLING	Moving Velocity	0.200337442
7	-10.56	2.86	0.8	-1	2019090118	LINGLING	Month	000000001000
8	-10.8	3.06	0.76	-0.88	2019090200	LINGLING	Location Longitude	0*12 1 0*23
9	-11.02	3.36	0.76	-0.88	2019090206	LINGLING	Location Latitude	00000000100
10	-11.14	3.64	0.7	-0.8	2019090212	LINGLING	History Direction (24 h)	00000010
11	-11.2	3.78	0.6	-0.68	2019090218	LINGLING	History Intensity Change (24 h)	0010
12	-11.22	3.92	0.5	-0.6	2019090300	LINGLING	Subtropical High	GPH

¹ **Details of $Data_{3d}$:** We crop data of 81*81 size from original ERA5 data to ensure that these data cover the 20°*20° region around the TC center.

² **Examples of $Data_{1d}$:** **ID** is the time step of TC, **LONG** is the Longitude (0.1°E) of TC, **LAT** is the Latitude (0.1°N) of TC, **PRES** is the minimum pressure (hPa) near the TC center, **WND** is the two-minute mean maximum sustained wind (MSW; m/s) near the TC center. **YYYYMMDDHH** is the date and time in UTC of TC, **Name** is the name of the TC.

³ **Environment data:** The environment data of TC LINGLING at the date 2019/09/06 18:00. 0*12 1 0*23 means the one-hot code includes 12 '0's, a '1', and 23 '0's.

Table 2: Details of TCN_D .

Table 2.1 presents the Meteorological Grid Data, which mainly includes information from several variables of ERA5. For each time step, we crop a 20°*20° area centered around the TCs current location. These data contain both temporal and spatial information, so we refer to it as $Data_{3d}$ (2D data with time series). A detailed explanation can be found on page 5, lines 318-342. We also show the description as follows:

There are many meteorological grid data about TC, like various variables in ERA5 dataset (ECMWF 2022). By leveraging this data, we can extract rich features to better represent the TC. For example, we can use the 500 hPa geopotential height data to describe the TC's pressure structure. The meteorological grid data are three-dimensional data (two-dimensional grid data with time series), hence referred to as $Data_{3d}$. In addition, $Data_{3d}$ encompasses not only geopotential height but also other grid data, like Sea Surface Temperature (SST), U-component of wind, and V-component of wind. All $Data_{3d}$ used in this study were collected from ERA5 reanalysis dataset. The details of $Data_{3d}$ are shown in Table 2.1.

After obtaining the $Data_{3d}$ data, we align them with $Data_{1d}$ based on the timestamp. We crop the data to cover 20°*20° region around the TC center. The spatial resolution is 0.25° and the temporal resolution is 6 hours. We collect geopotential height, U-component of wind, V-component of wind at 200, 500, 850 and 925 hPa pressure levels. Sea surface temperature data is also part of our $Data_{3d}$ set. Although there are some TC-related variables in our dataset, the TC is too complex to be fully represented by these data. Therefore, we will continue to update variables to meet various requirements of users and improve the dataset collaboratively.

Table 2.2 presents the Inherent Attributes Data, which primarily includes the TCs latitude, longitude, and intensity information (scalar variables with time series). We refer to this as $Data_{1d}$. The data displayed in the table has been normalized, and the specific normalization method is described in sub-section **Normalization of $Data_{1d}$** in the section **Data availability** on page 17, lines 1170-1194. Additional $Data_{1d}$ details are provided on page 5, lines 304-317, as shown in follows:

Inherent attribute data constitutes the foundational data in TCN_p , directly describing the location and intensity of TCs. It includes the data of longitude, latitude, pressure, and wind at or near the TC's center, collected from official TC records (Ying et al. 2014, Knapp et al. 2010), which is shown in Table 2.2. Additionally, these attributes serve as the target variables for prediction in our study. The inherent attribute data are one-dimensional data (scalar variables with time series), thus referred to as $Data_{1d}$.

The data in $Data_{1d}$ was normalized by some rules because normalized data is suitable for DL methods to extract useful information in them. All the rules are shown in sub-section **Normalization of $Data_{1d}$** in section **Data availability**.

Table 2.3 presents the Environment Data, which mainly includes additional data obtained through $Data_{1d}$ and $Data_{3d}$ analysis, such as the season in which the TC occurs, the subtropical high, and other related factors. Therefore, we refer to it as *Env-Data*. The data in the table has also been processed, and the specific processing methods and the meanings of the data are detailed in sub-section **Explanation of Environment Data** of the section **Data Availability** on page 17, lines 1195-1232. We have also added some descriptions on page 5, lines 343-372 in the manuscript, as shown in follows:

Environment factors, such as subtropical height, and seasonality, play a critical role in TC development, influencing the tendency of both track and intensity. Therefore, to enable deep learning models to leverage environmental information and enhance prediction accuracy, we collaborated with meteorologists to identify and extract crucial environmental data from both $Data_{1d}$ and $Data_{3d}$, which is shown in Table 2.3. This environment data, referred to as *Env-Data*, includes the movement velocity, month, the relative location on the earth (Location Longitude and Location Latitude), 24-hour history of moving direction, 24-hour history of intensity change, and subtropical high region. Some of these factors are often overlooked by previous deep learning methods.

Here, we also process the values of each variable in *Env-Data* to make them more suitable as input for deep learning models. The sub-section **Explanation of Environment Data** in section **Data availability** explains how we process the values. Specifically, we use the one-hot encoding of Location Longitude and Latitude to represent the TC's relative location on Earth, which differs from the location data in $Data_{1d}$. We aim to eliminate the numerical magnitude relationship inherent in latitude and longitude through this encoding method, as such relationships have no physical significance, particularly for longitude. This encoding approach allows the model to focus more on the relative position of the TC on Earth. Different relative positions correspond to distinct environmental conditions, allowing the model to learn the differences between them and, consequently, make more accurate predictions.

Additionally, we have updated Figure 1 and created Table 3 on page 8, as shown in follows, to describe the role of each variable in TC forecasting. All of this supplementary information will help readers better understand the data included in our dataset, the significance of each variable, and

make it easier to utilize our dataset.

Data Type	Data Name	Contribution for TC forecasting
Data_{1d}	LONG	The target attribution to be predicted
	LAT	The target attribution to be predicted
	PRES	The target attribution to be predicted
	WND	The target attribution to be predicted
Data_{3d}	Sea surface temperature	TCs mainly intensify by absorbing heat energy from the ocean. Warm sea surface temperatures (typically above 26.5°C) provide the energy that drives the development and strengthening of typhoons. High-temperature ocean regions promote water vapor evaporation, increasing the moisture supply to the TC and further enhancing its strength. When a TC passes over warm ocean waters, the higher the sea surface temperature, the more likely the typhoon is to intensify. Conversely, when a TC moves into cooler ocean regions, its intensity may weaken. SST data can help predict changes in the typhoon's intensity.
	Geopotential Height	As a powerful low-pressure system, the strength of a TC is closely related to the surrounding geopotential height field. Strong TC exhibit lower geopotential heights at various atmospheric levels, where this is more pronounced. By monitoring the geopotential heights at these levels, the intensity trend of a TC can be assessed.
	U-component of wind and V-component of wind	The U-component (zonal wind) and V-component (meridional wind) together describe the horizontal wind structure in a three-dimensional wind field. By analyzing the UV components, the wind speed and direction around a TC can be accurately characterized, providing insight into the TC's convective structure, airflow patterns, and surrounding environment. This is crucial for assessing the intensity and wind field variations of the TC. The TC's path is typically guided by large-scale atmospheric circulation, and these steering currents can be captured through the UV components. By analyzing the U and V components at different atmospheric levels, especially at the 500 hPa level, the direction of the steering flow can be determined, helping to predict the future path of the TC.
Env-Data	Moving Velocity	The moving velocity of a TC directly influences the prediction of its future path. By understanding its current speed, it is possible to predict the geographical location it will reach at a given time. The moving velocity of a TC is closely related to changes in its intensity. Generally, slower-moving TCs remain over warm ocean areas for longer periods, providing more opportunity to absorb energy, which may lead to intensification. In contrast, faster-moving TCs may weaken more quickly when interacting with cold air or land. Therefore, the moving velocity helps in estimating the intensity change trends of a TC.
	Month	The occurrence of TCs follows a distinct seasonal pattern, typically being most active during the summer and autumn months. By understanding the month, it is possible to better predict the likelihood and frequency of TC formation. Seasonal changes in atmospheric circulation during different months can lead to variations in TC track patterns. For instance, in the Northwest Pacific, when the subtropical high is strong during the summer, TCs may move westward, affecting East Asia. However, in autumn, as the subtropical high shifts eastward, TCs are more likely to move northward or northeastward.
	Location Longitude and Location Latitude	Different oceanic and atmospheric conditions in various regions affect the formation and development of TCs. For example, regions like the Northwest Pacific and the South China Sea, where the sea surface temperature is consistently warm, are more conducive to the formation and intensification of typhoons. In contrast, colder regions, such as higher latitudes, are typically unfavorable for the formation and sustenance of TCs. Each region has its unique patterns of typhoon activity. By analyzing historical data, it becomes easier to predict the behavior of current typhoons. For instance, in certain regions, the paths, formation frequency, and intensity of TCs follow specific patterns and trends, and utilizing these historical data can improve prediction accuracy.
	History Direction (24 h)	By analyzing the TC's movement direction over the past 24 hours, the model can better assess the future trajectory trends of the TC. TC movement direction typically does not undergo drastic changes in a short period, making historical direction a reliable basis for predicting future paths. The TC's movement is guided by the surrounding atmospheric circulation, such as the position of the subtropical high or troughs. Changes in the historical direction reflect the influence of these steering flows, helping to predict whether the TC will deviate towards a particular direction.
	History Intensity Change (24 h)	By analyzing the TCs intensity change over the past 24 hours, the model can help determine the future trend of intensity changes. If the TC has rapidly intensified or weakened in the past 24 hours, it may indicate that this trend will continue, unless significant changes occur in the external environment. The historical intensity change reflects the influence of the surrounding environmental conditions, such as sea surface temperature and wind shear. For example, if a TC has rapidly intensified over the past 24 hours, it may suggest that it has passed through an area with high sea surface temperatures or low wind shear, and this information can help assess future intensity changes.
	Subtropical High	The subtropical high is one of the most important weather systems influencing the path of TCs. It is typically located in the subtropical region (around 20° to 30° latitude) and is a large high-pressure system. The path of a typhoon often moves along the edge of the subtropical high, guided by its steering flow. Generally, the stronger the subtropical high, the more likely the TC will move westward along its southern edge; when the subtropical high weakens or shifts eastward, the TC may move northward or northeastward. Although the subtropical high primarily influences the path of TCs, it can also affect their intensity to some extent. The subtropical high is usually associated with a stable atmospheric environment and weak wind shear, which are favorable conditions for the TC to maintain or strengthen. When a TC passes through a region under the control of a strong subtropical high, it is likely to maintain its intensity or even strengthen.

Table 3: Data's contribution for TC forecasting.

We have also addressed and revised other important comments one by one for reviewers' questions. The reviewers' comments will be **highlighted in bold and blue text**, while our responses will be in

regular font. Any quotes from my manuscript will be presented in a regular font and underlined. The responses to Reviewer 1 can be found on pages 6-10 of this letter, and the responses to Reviewer 2 start from page 10.

For Reviewer #1

Question #1: The authors could train and test their model for longer time periods, which is considered essential for better emergency management.

Answer #1: Thanks for the constructive suggestion. 24 h forecast may be not long enough for government and people to prepare for the coming TC. Thus, we obtain the forecasting performance of our model TCN_M from 6 h to 72 h and compare it with one of previous best methods MMSTN and China Meteorological Observation method CMO. We show results in sub-figure 5.b-1,b-2, and b-3, on page 10. We also show the sub-figure as follows:

We also analyze the results in the table and add some descriptions on page 11, lines 635-659. The description is as follows:

We also aim to explore the longer time forecasting performance of our model. Thus, we obtain the forecasting performance of our model from 6 to 72 hours and compare it with one of previous best methods -- MMSTN and China Meteorological Observation method CMO, which is shown in Fig. 5.b-1, b-2, and b-3. From the results in the table, the proposed TCN_M outperforms MMSTN and keeps the top rank in the task of TC intensity prediction. Our method achieves the best performance in track prediction for the 6-hour and 12-hour forecasts. However, after 12 hours, its performance begins to fall behind CMO, and the gap continues to widen as the prediction time increases.

We believe that the reasons of our model under-performing in long-term tropical cyclone (TC) track prediction are mainly attributed to the accumulated errors in multi-step forecasting. Intensity prediction performs better than the track prediction, possibly because track prediction involves forecasting both latitude and longitude, which are two dimensions. As the number of prediction steps increases, the accumulated errors grows exponentially. In contrast, intensity prediction involves only a single dimension, making it easier to predict. Although accumulated errors still exist, they grow linearly. Therefore, intensity prediction outperforms track prediction.

Question #2: In lines 313 and 314, the authors mentioned, environmental data was extracted from the 1D and 2D datasets. As the 1D and 2D datasets already contain the environmental

properties used in this work, why were those properties extracted and then used again in the prediction process?

Answer #2: Here, I would like to first clarify the modified description about our dataset in the revised version. In this version, we modify all '2D data' to '3D data' as '2D data' may confuse readers, which is explained on page 5, lines 311-312 and 323-325.

We are sorry that we do not clarify what kind of specific data in 1D (scalar variable with time series), 3D (2D grid data with time series) and environmental dataset. Thus, we make a table to show the details of those three data. 1D, 3D and environmental dataset are shown in Table 2. From the details in these tables, we could find the data in environmental dataset is different from that in 1D and 3D datasets. 1D dataset only includes normal and basic information about TC, like the track and the intensity. The 1D data in environmental dataset include much information, like the Moving Velocity, Month, History Direction (24 h).

Question #3: Whether the variables were considered for a single level or from multiple vertically arranged pressure levels is not mentioned.

Answer #3: When training the model TCN_M , we considered the **single level variables**. Considering the cost of computer resources, we input single pressure level data to our model, which just costs less one day to train and several seconds to inference with an A6000 GPU (48G). This resource requirement is affordable for most research groups that are interested in AI for meteorology. However, in our dataset TCN_D , we provide various meteorological variables on **multiple pressure level**, which is shown Table 2.1, to meet different requirements from researchers.

Question #4: However, they could improve it through linking the datasets with TC properties in methodology. In particular, the authors could mention which type of data is serving which purpose in the developed framework. How 1D and 2D datasets contributed to TC track and intensity prediction appears to be an important thing for the readers to understand.

Answer #4: It is a critical advice to improve our paper. We add a Table 3, on page 8, to introduce why we choose those data to build the TC dataset and how those data contributed to TC track and intensity prediction. Readers will gain a better understanding of the motivation behind the creation of our dataset and the role each type of data plays in TC prediction. This will make people more inclined and find it easier to use our dataset.

Question #5: 1. Whether the datasets were arranged sequentially following the TC formation time and lifespan. Whether TC lifespans were divided to prepare train and test datasets. 2. Why was data collected between 1950 and 2016 used for both training and testing? Why was data collected during 2017 and 2021 used only for testing?

Answer #5: We are sorry that we did not provide more detail information about dataset arrangement and division.

1. No, the dataset was not arranged sequentially based on TC formation time and lifespan. For the data from 1950 to 2016, the training (80%) and validation (20%) sets were randomly split to ensure diversity and randomness in the training process. The TC lifespans were not divided; instead, the entire lifespan of each TC was assigned to either the training set or validation set to prevent data leakage and ensure independence between the two sets. However, all data from

2017 to 2021 was reserved as the final testing set to evaluate the model's performance on unseen and more recent TCs.

- The data from 1950 to 2016 was split into training and validation sets to provide a diverse and extensive dataset for model training and optimization. This period includes a wide variety of TCs, which helps the model learn general patterns and relationships. In contrast, the data from 2017 to 2021 was reserved exclusively for testing because it represents more recent and unseen TCs. This setup allows for an unbiased evaluation of the model's generalization performance on new data and ensures that the testing set reflects the model's ability to handle future predictions in a realistic scenario.

Question #6: What was the performance of the method in case of rapid intensifying and recurving TCs? The authors may add this information, as these TCs pose extra complexity in forecasting end and threat in emergency management end.

Answer #6:

Following the reviewer's suggestion, we analysis the rapid intensifying and recurving TC cases. We show the results of these cases in figure 5, on page 10. We also show the sub-figure as follows:

Sub-figures (c-1), (c-2), and (c-3) illustrate the comparison of TCN_M's performance on cases with rapid changes in intensity or track against its overall performance on the entire dataset. In these sub-figures, two solid lines in the represent the results of CMO (an official meteorological agency from China) and TCN_M on the complete Western North Pacific (WP) test set, respectively. The comparative results are based on WP sea area data, as CMO only provide performance results for the WP region. CMO- r_{Int} and TCN_M- r_{Int} represent the results of CMO and TCN_M on rapid intensifying TC cases, respectively. CMO- r_{Tra} and TCN_M- r_{Tra} represent the results of CMO and TCN_M on curving TC cases, respectively.

We also give more detail descriptions on pages 11-12, lines 660-733 to analysis the results, which are as follows:

We also conducted experiments to analyze the performance of our method on rapid intensifying and recurving TC cases, as shown in Fig. 5.c-1, c-2, and c-3. First, we focus on the selection of rapid intensifying cases. Rapid intensification is currently defined as a tropical cyclone (TC) experiencing an increase in maximum sustained wind speed near its center by at least 15.432 m/s within 24 hours. A typical case is defined as a TC sequence spanning 12 time steps, with a time resolution of 6 hours, resulting in a total length of 66 hours. Of these, the first 42 hours represent the observed TC, while the remaining 24 hours are the prediction target. We consider a 48-hour time window, which includes 24 hours before and after the current time step. If, at any point within this 48-hour window,

the maximum wind speed near the TC's center increases by at least 15.432 m/s over a 24-hour period, we classify the case as a rapid intensifying TC case. In the WP test dataset, there are 2602 cases in total, 192 of which are rapid intensifying TC cases, accounting for 7.37%.

For recurving TC cases, there is no established standard for defining such cases. We set a reasonable criterion to select these cases. Similarly, we use a 48-hour time window encompassing 24 hours before and after the current time step. Within this window, we calculate the angular change with the TC track every 6 hours. If the track experiences at least one angular change of over 70 degrees within any 6-hour interval during this window, we classify the case as a recurving TC case. Based on our analysis, there are 279 recurving TC cases, representing 10.72% of all WP cases.

The experimental results reveal an interesting phenomenon. In cases of rapid intensification of tropical cyclones, the performance of CMO shows a more significant decline in predicting intensity (pressure and wind) compared to its performance on the entire test set. In contrast, although our model also experiences a decline, the magnitude of this decline is smaller than that of CMO. The decline of our model can be attributed to the complexity of sudden changes in TC intensity and the smaller proportion of relevant samples in the training set. These factors make it challenging for the model to simulate the rapid intensification process. However, for track prediction, our model demonstrates significant improvement compared to the results from the entire test set. The average rate of change in the track direction in rapid intensifying TC cases is 10.1 degrees per 6 hours, and the maximum sustained wind speed near the center is 44.96 m/s. This suggests that rapid intensification often occurs when the TC is already a strong system with a well-developed structure and high intensity. At this stage, the TC's track is less influenced by environmental factors, resulting in nearly linear motion, which is easier to predict, with an average change rate of 10.1 degrees per 6 hours. As a result, the model performs well in predicting the track for these cases.

A similar phenomenon is observed in recurving TC cases. The CMO method also shows a significant decline in track prediction performance compared to its performance on the entire test set. In contrast, although our model experiences a decrease in track prediction, the drop is less pronounced than that of CMO and our model demonstrates substantial improvement in intensity prediction compared to its performance across the entire test set. We further analyzed this phenomenon. In recurving TC cases, the average change rate of maximum sustained wind speed near the center is 3.59 m/s per 24 hours, and the average TC intensity is 22.21 m/s. The smaller intensity changes in these cases contribute to the model's improved intensity prediction performance. These interesting phenomena could be considered in future work. For instance, rapid intensification is more likely to occur during the high-intensity stage of a TC, whereas track direction changes are more likely during the low-intensity stage.

Question #7: The authors could cite a few more works related to TC meteorology.

Answer #7: Thank you for your suggestion. TC meteorology is crucial for AI-based TC prediction. Therefore, to improve this paper and provide readers with a better understanding of the context of our research, we have reviewed and cited the following references related to TC meteorology in our manuscript on page 1, lines 43-45 and 50-51, in addition to the existing relevant TC meteorology citations.

- Emanuel, K. Increasing destructiveness of tropical cyclones over the past 30 years. *Nature* **436**, 686–688 (2005). <https://doi.org/10.1038/nature03906>
- Wang, G., Wu, L., Mei, W. *et al.* Ocean currents show global intensification of weak tropical cyclones. *Nature* **611**, 496–500 (2022). <https://doi.org/10.1038/s41586-022-05326-4>
- Emanuel, K. "Will Global Warming Make Hurricane Forecasting More Difficult?". *Bulletin of the American Meteorological Society* 98.3 (2017): 495-501. <https://doi.org/10.1175/BAMS-D-16-0134.1>

For Reviewer #2

Question #1: Indeed, the database is useful but, it is rather similar to previous studies (which might not have released it), so it is just a matter of re-extracting reanalysis using IBtracs catalog: useful, but not novel.

Answer #1:

We appreciated that reviewers acknowledge the usefulness of the database. Compared to previous studies, some of which also utilize reanalysis data and IBTrACS to train their models, however, our dataset offers broader temporal coverage (more than 70 years), and encompasses a wider range of ocean basins (6 ocean basins). Additionally, our dataset provides richer and more diverse data, including environmental data, which is rarely incorporated in previous deep-learning approaches. Details of environmental data are presented in Table 2.2 on page 7. These environmental data is an important part of understanding and predicting TCs. However, these types of environmental data are complex and difficult to use directly. Our dataset has preprocessed this data to make it more suitable for deep learning models. These environmental data also play a crucial role in enabling our model to learn and model TC effectively. The comparisons between the second and fourth rows, as well as the third and fifth rows, in the ablation study (Table 4.1, page 13) demonstrate that both the environmental data and the environment-specific network designed for this data significantly contribute to improving the model's performance.

	H	M-GC	Env-T	TCN _D	Track (km)				Wind (m/s)			
					6 h	12 h	18 h	24 h	6 h	12 h	18 h	24 h
1					27.52	59.93	97.28	139.34	0.85	1.42	1.97	2.47
2	✓				25.80	50.08	75.89	105.53	0.81	1.36	1.86	2.23
3	✓	✓			24.39	47.50	73.76	106.08	0.73	1.24	1.72	2.17
4	✓		✓		25.02	48.50	72.89	102.28	0.85	1.36	1.77	2.07
5	✓	✓	✓		23.84	45.03	67.93	95.39	0.72	1.15	1.48	1.81
6	✓	✓	✓	✓	22.98	43.83	66.41	93.76	0.72	1.09	1.43	1.75

Here, (H) represents Receiving Heterogeneous Meteorological Data, (M-GC) represents Multi-Generators with Generator Chooser Network, and (Env-T) represents Environment-Time Network. The item of TCN_D means that the results were from our model trained on all the six sea regions data.

What is more, as reviewer mentioned, previous studies have utilized reanalyzed data and TC best track data; however, they did not release their datasets or code. This lack of transparency makes it

challenging for subsequent researchers to replicate their methods or to discover prior work, which hinders progress in the field. Our goal is to enable future researchers to focus on model development and the analysis of TC-related data, thereby alleviating the burdens of data collection and standardization. This approach not only saves time for the development of new methodologies but also attracts new researchers to engage in this important task. Thus, we have open the test code of our model on <https://github.com/xiaochengfuhuo/TropiCycloneNet> and our dataset on <https://github.com/xiaochengfuhuo/TropiCycloneNet-Dataset> for the review purposes. After that this work was accepted, we will also open our complete training code of our model and update a more detailed documentation on GitHub to facilitate user accessibility.

We also compared our dataset with those of previously published tropical cyclone datasets, like China Meteorological Administration Tropical Cyclone Best Track Data (CMA-BST), Tropical cyclones Dataset based on satellite cloud images (TCTSCI), International Best Track Archive for Climate Stewardship (IBTrACS), and Satellite Image Dataset for Tropical Cyclones (Digital Typhoon), as shown in Table 1 on page 4:

Dataset Contents	CMA-BST	TCTSCI	IBTrACS	Digital Typhoon	TCN _D
Data_{1d}	✓	✓	✓	✓	✓
Data_{3d}	✗	✓	✗	✓	✓
Env-Data	✗	✗	✗	✗	✓
Global	✗	✗	✓	✗	✓

It is a comparison of TCN_D and other well-known TC dataset. $Data_{1d}$ represents inherent attributes data. $Data_{3d}$ refers to various meteorological grid data. *Env-Data* means environment data. Global indicates whether the dataset encompasses nearly all sea areas on this planet. We also add some description to explain this table, as shown on page 2, lines 111-131:

In the field of TC prediction, there are several important datasets, such as International Best Track Archive for Climate Stewardship (IBTrACS) (Knapp et al. 2010), Tropical cyclones Dataset based on satellite cloud images (TCTSCI) (Huang et al. 2021), Satellite Image Dataset for Tropical Cyclones (Digital Typhoon) (Kitamoto et al. 2023), and China Meteorological Administration Tropical Cyclone Best Track Data (CMA-BST) (Ying et al. 2014; Lu et al. 2021). CMA-BST includes only inherent attribute data, such as the track and intensity of TCs. Compared to CMA-BST, TCTSCI and Digital Typhoon contains meteorological grid data, such as satellite images and reanalysis data; however, both datasets are limited to TCs in the Western North Pacific region. In contrast, IBTrACS is a global best track dataset for TCs, but it only includes inherent attribute data. Furthermore, these datasets neglect some important environmental factors of TCs, which play a critical role in TCs' development. Thus, these datasets do not provide sufficient information for deep learning models to effectively learn and predict TC. That is, there is still no open large-scale, multi-modal dataset for TC prediction, which hinders its development.

For novelty, when we read this paper, we may think that the idea of building a new open large TC dataset is interesting. *After* we know what is included in this dataset, we might not feel surprise because most data in this dataset is widely used and the motivation of choosing these type of data

(IBTrACS and ERA5 reanalysis data) is obvious. The novelty, however, must be evaluated *before* the idea existed. The inventive novelty was to have the idea in the first place. If an idea is easy to explain and obvious in hindsight, it does not contradict the fact that it is novel.

Question #2: The formatting of the data is very important for a database to become a proper benchmark while it is not described, and the many important information are not explained (the size of the 2D data, the list of the proper fields, etc.) and the data is not released yet.

Answer #2: We are sorry for that we did not provide enough details about dataset. It is not benefit for readers to understand and use our dataset. Review's advice is really help for us to improve this paper. Thus, we make a table to show the details of 1D (scalar variable with time series), 3D (2D grid data with time series, '2D data' may confuse readers so we modify it to 3D data, which is explained on page 5, lines 311-312 and 323-325.) and environmental dataset, which are shown in Table 2. The size of the 3D data is explained on page 5, lines 332-334:

We crop the data covering $20^{\circ} \times 20^{\circ}$ region around the TC center. The spatial resolution of them is 0.25° and the time resolution is 6 hours.

Additionally, we have released the TCN_D dataset on GitHub. To facilitate the review process, we provide both the full dataset and the separate test dataset for download. For more details, please visit: <https://github.com/xiaochengfuhuo/TropiCycloneNet-Dataset>. A more detailed document will be updated on GitHub once this work is accepted.

Question #3: What you call 'Environmental' input information is not well defined, and can be very confusing. In particular, the sentence 'Previous deep learning methods often overlook the impact of environment-related factors, which are critical in the development of TC' is not clear, what do you mean by that? Also, it is not clear the link between env. data and the 1d / 2d data used in the other branches of your model. Please explain.

Answer #3: Thank you for your feedback. To better illustrate our data, we have created a table that provides more detailed information about our dataset. Table 2.3 presents the specifics of the environmental data. By comparing the three sub-tables, the differences among the three types of data—1D, 3D (2D+time), and environmental—can be more clearly understood.

As for the sentence mentioned by reviewer in this question, it refers to the significant influence that environment-related factors have on the development of TCs. If our model can learn how environmental factors affect TC development from the environmental data, it will help the model produce more accurate predictions. We have also supplemented our manuscript with additional descriptions to clarify this point.

Previous deep learning methods often overlook the impact of environment-related factors, while these factors are critical in understanding the development of TC

Question #4: When talking about the 'Data 1d', it is slightly misleading (not in 1 dimension but in 0, no?) - or it is 1d if considered time series (but in that case the others are 2D+t)?

Answer #4: Thank you for your insightful feedback regarding the term ' $Data_{1d}$ '. It is misleading as

it implies one-dimensionality, while it may be more appropriate to consider it as a zero-dimensional dataset or as a time series. Thus, we would like to modify ' $Data_{2d}$ ' to ' $Data_{3d}$ ' (2D+t) and keep ' $Data_{1d}$ ' unchanged. I will revise the relevant descriptions to clarify this distinction and will include additional explanations to enhance understanding. The descriptions added on page 5, lines 311-312 and 323-325 are as follows:

The inherent attribute data are one-dimensional data (scalar variables with time series), thus referred to $Data_{1d}$.

The meteorological grid data are three-dimensional data (two-dimensional grid data with time series), hence referred to $Data_{3d}$.

Question #5: Meteorological data : please detail, I guess you are meaning that you are using reanalysis, but give details about the database (ERA5?). Are these data available in near real-time? Could your model be thus used in an operational way?

Answer #5: The primary source of meteorological data is the ERA5 reanalysis dataset. More information is shown in Table 2.1. For the 'near real-time' concern, Data 1d and Environmental data (except ERA5 data) could be obtained in near real-time. With collaboration from ECMWF, the data related to ERA5 reanalysis data can also be obtained in near real-time. For example, Pangu Weather also use ERA5 reanalysis data and ECMWF have applied Pangu Weather model in their system to execute the real-time prediction [1]. Thus, we think the proposed model could be used in an operational way. So, using reanalysis data in research is acceptable.

Question #6: Please clarify which features, how many time steps, what is the size of the window, if it is centered on the TC (if so it means the window moves with time)? Do you think that a fixed window would be better?

Answer #6: The variables (features) in TCN_D are shown in Table 2. The time steps of our input are 8 and time steps of output are 4. The time resolution is 6 hours. That is, we predict future 24 hours data of TC track and intensity. For all the reanalysis data in TCN_D , we extracted a **20-degree by 20-degree** matrix from global reanalysis data, with the center of this matrix corresponding to the current TC's center. The window moves with time. We also add some descriptions in the paper to clarify these questions in page 4, lines 270-274 and page 5, lines 332-334, respectively:

In this work, we set $n = 8$ and $m = 4$, which means that we input 42 hours historical TC data into model and the model can give future 24 hours TC track and intensity predictions.

We crop the data covering $20^{\circ} \times 20^{\circ}$ region around the TC center. The spatial resolution of them is 0.25° and the time resolution is 6 hours.

Fixing the window location is an interesting idea. However, it may raise several questions:

Firstly, a fixed window must encompass all potential TC locations during the timeframe. Expanding the current $20^{\circ} \times 20^{\circ}$ range may improve environmental coverage but increases computational demands and risks diluting the focus on TC-specific features, requiring careful balance.

Secondly, the criteria for determining the window's size and position need further study. For global TC forecasting, a global-scale window is the simplest choice but raises computational challenges and questions about integrating global and localized information.

In summary, while the current $20^{\circ} \times 20^{\circ}$ window around the TC center is a reasonable choice, the reviewer's suggestion opens new possibilities worth exploring. It requires further investigation and may serve as future research directions.

Question #7: Why did you use old data, i.e. prior to 2000, since the reanalysis and also the 1d data is most probably of bad quality?

Answer #7: Thank you for your question regarding the use of older data, specifically prior to 2000. While it is true that such data may have quality issues, including noise, deep learning methods are designed to effectively learn from large datasets. A larger dataset, even with noise, provides more opportunities for the model to learn robust patterns, ultimately enhancing generalization and performance while preventing overfitting and improving the stability of prediction results [2].

Question #8: Why don't you use any data after 2021?

Answer #8: Building an open large and high-quality TC dataset started in 2022. At that time, one of our goals is to collect rich TC data before 2022. After we have built a basic TC dataset, we started the research of designing DL model and neglected that there are new TC data with the time goes by. Your review inspired us, the data of 2022 and 2023 have released on GitHub. In the experiment of this paper, we would like keep the test set as set as before for fair comparison, as some of comparing DL method do not provide the results on 2022 and 2023.

Question #9: State of the art: MMSTN is not part of your state-of-the-art section while you are using to compare later on. Please add it and explain what it is. Also, by looking at the original MMSTN paper, it does not seem to be the best in TC tracking. So I would suggest comparing with other models. More generally, it is not clear why you chose to compare against these architectures and not others? What do each of them use as input?

Answer #9: Follow reviewers' advice, we will add the information mentioned in the question to the **Comparison with State-of-the-art Deep Learning Methods** section on page 7, lines 473-516, which including the brief introduction of these deep learning methods, why we select them, and the data they used. The descriptions we added are shown as follows:

To comprehensively compare TCN_M with other deep learning methods, we selected nine methods, categorized into three groups for comparison. The first group consists of classic time-series prediction methods, such as LSTM (Hochreiter and Schmidhuber 1997), GRU (Cho et al. 2014), and SGAN (Gupta et al. 2018). The second group includes modifications of time-series networks tailored for tropical cyclone (TC) prediction, such as NMPT (Gao et al. 2018), DLM (Pan, Xu, and Shi 2019), GBRNN (Alemany et al.2019), and MMSTN (Huang et al. 2022). The third group

includes large-scale meteorological models, including Pangu-Weather (Bi et al. 2023) and Fengwu (Chen et al. 2023).

LSTM, GRU, and SGAN are classic time-series prediction methods in computer science. All of these methods are open-source, but they only accept one-dimensional data as input. Therefore, we used one-dimensional data as input for these methods, which includes both track and intensity information.

NMPT, DLM, and GBRNN are modifications of time-series networks, such as RNN and LSTM, adapted by previous researchers for TC prediction. These methods are not open-source, but since they use only one-dimensional data and they are relatively simple. We reproduced these models based on their descriptions in the papers and applied them to our dataset. MMSTN is a method among them with open-source code and data (the one-dimensional best-track dataset). It also predicts track, maximum wind speed near the center, and minimum central pressure simultaneously. So we used it as the main comparison method.

Pangu-Weather and Fengwu are designed for global meteorological variable forecasting, such as mean sea level pressure and various variables at different pressure levels. TC track and intensity prediction is not the direct goal of these large models but can be achieved by calculating the meteorological variables they predict. It is worthwhile to compare our model with these large-scale models. The data used by these large-scale models primarily include several variables from ERA5 reanalysis data: geopotential height, specific humidity, temperature, U-wind, and V-wind at pressure levels [50, 100, 150, 200, 250, 300, 400, 500, 600, 700, 850, 925, 1000], as well as surface-level atmospheric data such as 10m U-wind, 10m V-wind, 2m temperature, and mean sea level pressure.

Additionally, in the original MMSTN paper, the MMSTN model was considered the best method among the deep learning approaches. However, its performance in track prediction is inferior to the China Meteorological Administration's (CMA) CMO method. In Table 3-2, we also compared the prediction performance of our model with CMO in the WP region. For predictions beyond 12 hours, a performance gap still exists between our method and CMO.

Question #10: Did you compare with an architecture using 2D data?

Answer #10: Sorry for that we did not select a suitable methods using 3D (2D+t) data. We tried to find open-source code and data, but unfortunately, we couldn't. We also attempted to reproduce these methods, but it's challenging to fully replicate them just based on the descriptions in the papers. The only way to compare these models is to retrain one of them and compare it on the same years tested in the respective papers. However, different methods test on different years, so we chose one method for comparison. The comparison results are shown as follow table:

Methods	Distance (km)			
	6 h	12 h	18 h	24 h
DBF-Net	31.30	58.94	87.60	119.05
TCN _M	23.55	43.16	64.43	90.79

Due to the limitation on the number of tables allowed in the main text of Nature Communications,

we are unable to include this table in the manuscript. However, we will make this table along with its relevant descriptions available on the GitHub repository associated with this work. This table shows the Comparisons of average absolute error of TC track prediction of different methods from the year 2014 to 2017. DBF-Net [3] is a multi-modal method with 3D data (2D grid data with time series). It uses the data of the TC best track dataset and GPH to make a prediction. However, DBF-Net is experimented from the year 2014 to 2017 while our method TCN_M is experimented from the year 2017 to 2021. Thus, we cannot directly compare the results of TCN_M with that of DBF-Net. We want to get the results of DBF-Net from the year 2017 to 2021, but the code is not open momentarily. So, we train and verify TCN_M on the data from the year 1950 to 2013 and obtain the test results from the year 2014 to 2017.

During finding suitable TC-related work to compare, we identified several issues: the model codes are not open-source, and there is no standardized, open-source dataset for model validation. Before starting their research on TC prediction, most researchers seem to be repeating the same task, which is data collection. After training their models, they often select different regions and different years of TC data for prediction. This lack of a unified benchmark in the field hinders progress. Therefore, one of the reasons we created this dataset and open-sourced our method is to address this gap and contribute to the development of the field.

Question #11: Did you compare with recent meteorological forecasting methods like GraphCast that you describe in your introduction?

Answer #11: Thank you for your advice. Meteorological large-scale models, like GraphCast, Pangu-weather, and Fengwu, have demonstrated strong capabilities in forecasting global weather variables. Currently, we found that some of these large-scale models have released part of their code related to global predictions, though GraphCast has not yet open-sourced its TC prediction code. Furthermore, as seen in <https://sites.research.google/weatherbench/>, the prediction capabilities of GraphCast and Pangu-weather are quite similar, which is shown as follows:

[REDACTED]

Therefore, we chose to compare our model with Pangu-weather and FengWu. The results are shown

in Figure 5.a-1, a-2 and a-3:

The results of Pangu-weather and FengWu are shown in the red boxes. For TC wind prediction, Pangu-weather and FengWu do not provide the prediction of maximum wind speed near the TC center and the variables predicted from them cannot be used to calculate TC maximum wind speed. We also add some descriptions to show analysis of the results, which is shown on pages 9-10, lines 544-584:

We are also interested in the performance of large-scale models. The results of Pangu-Weather and Fengwu are shown in Fig. 5.a-1 and a-2. The experimental results show that large-scale models have a significant advantage in TC track prediction, as shown in sub-figure 5.a-1. However, when predicting TC intensity, both of these large models seem to struggle to provide accurate predictions, as shown in the top part of the sub-figure 5.a-2. The reasons are as follows:

- **Reasons for the good track performance of large models:** TC tracks are primarily influenced by large-scale atmospheric flows, such as the subtropical high and wind fields. These atmospheric variables are relatively stable and easier to simulate and predict with meteorological models. The Pangu-Weather model effectively captures these large-scale dynamic structures, leading to its strong performance in track prediction.
- **Reasons for poor intensity performance of large models:** Pangu is a global-scale weather prediction model. On a global scale, the evolution of TC intensity is a relatively small-scale process, which may require region-specific models focused on TC prediction for accurate capture the evolution of TC intensity. Therefore, specialized datasets and methods tailored to TC prediction are necessary for better intensity forecasting.
- **Reasons for our model's better intensity prediction:** Our model not only extracts features from 3D reanalysis data but also learns from direct one-dimensional historical information, such as the TC's past track and intensity. This enables the model to learn how TCs evolve over time. Additionally, our environment module extracts features from historical track and intensity changes within the environmental data of our dataset, allowing the model to capture more patterns of TC behavior over time. Due to the high computational cost, Pangu uses only current data to predict future values without incorporating historical data. Fengwu uses data from the current and past six hours (two time steps) to predict future outcomes. In contrast, our model learns from a total of eight time steps, covering 42 hours of historical data, allowing it to extract more information.

Question #12: Have you considered using a self-supervised task in order to improve the extraction of reanalysis features (since the TC data time steps is rather small)?

Answer #12: We have not employed a self-supervised task in our current work. One of the main reasons is that our focus has been on leveraging domain-specific (TC) information and the relationships between different factors in reanalysis data to guide feature extraction, which aligns well with the task requirements. Additionally, incorporating a self-supervised approach would require significant adjustments to our model architecture and training pipeline, which is quite different from this study.

That said, we acknowledge the potential of self-supervised learning to enhance feature extraction, particularly given the relatively small time steps in TC data. Exploring self-supervised techniques to improve reanalysis feature representation is a promising direction, and we plan to consider it in future work.

Question #13: The data2D branch: the 2 steps and the 2 arrows (in the schema, the En-LSTM and the red embedding) are not clear. Explain how is made the 'embedding' after the Unet, and what is passed in the En-LSTM branch. Why do you need to input it also in the blue embedding network, and how do you input 2D features with the trajectory/intensity data?

Answer #13:

(1) Explain how is made the 'embedding' after the Unet

Sorry for that we did not provide more information to show how my model works, which makes reviewers confuse. Thus, we updated Figure 2, which is shown as follows:

Figure 2: TropiCycloneNet framework. It includes TropiCycloneNet Dataset (TCN_D) and TropiCycloneNet Model (TCN_M). TCN_D includes various data with time series. For TCN_M , the golden branch is the environment data (Env -Data) encoder, called Environment-Time Network (Env-T-Net). The blue branch is the inherent attributes data of TC ($Data_{1d}$) encoder, called 1D-Data Encoder. The red branch is the meteorological grid data ($Data_{3d}$) encoder, called 3D-Data Encoder. The **Roulette** selects different generators by the probability array, the circular ring below GC-Net, from Generator Chooser Net. All predictions from selected generators constitute the multiple potential tendencies of TC. Best viewed in color.

In the ' $Data_{3d}$ Branch' we input the two-dimensional TC data from n historical time steps into a 3D U-Net model, which outputs the two-dimensional TC data for m future time steps. To encode the two-dimensional TC data for both historical and future time steps, we use a multi-layer perceptron (MLP) to embed the two-dimensional data at each time step into a feature vector of length d , which is shown as the blue (e_{3d}^{En}) and red (e_{3d}^{De}) squares in the figure. In addition to the modifications to ' $Data_{3d}$ Branch' in the figure, we also revised the corresponding equations and

their descriptions. Modified Equation (1) and (2) and corresponding descriptions in page 14, lines 911-918, are as follows:

$$\hat{\mathbf{Y}}_{3d} = 3DUnet(\mathbf{X}_{3d}; W_{3D-Unet}) \quad (1)$$

$$e_{3d}^{En}, e_{3d}^{De} = \phi([\mathbf{X}_{3d}, \hat{\mathbf{Y}}_{3d}]; W_{MLP_{3d}}) \quad (2)$$

where \mathbf{X}_{3d} represents the history sequence $Data_{3d}$ of TC, $\hat{\mathbf{Y}}_{3d}$ represents the future predicted $Data_{3d}$ of TC, and e_{3d}^{En} represents the $Data_{3d}$ feature input for the 1D-Data Encoder, e_{3d}^{De} represents the predicted $Data_{3d}$ feature input for Generators. $W_{3D-Unet}$ and $W_{MLP_{3d}}$ represent the weights of the 3D-Unet and MLP respectively. Thus, the feature extraction of $Data_{3d}$ is completed, resulting in features e_{3d}^{En} and e_{3d}^{De} .

(2) What is passed in the En-LSYTM branch? Why do you need to input it also in the blue embedding network?

In the ' $Data_{3d}$ Branch' our module only extracts spatial features from the data, compressing the spatial dimension to 1D while retaining the temporal dimension. This indicates that the module does not extract temporal features from the 3D data. Therefore, we input this into the blue embedding network to fuse e_{3d}^{En} with the feature of $Data_{1d}$, and then feed the combined feature e^{En} into the En-LSTM to extract temporal features, as mentioned in page 14, lines 919-928. This approach effectively reduces model complexity and improves training speed.

(3) How do you input 2D features with the trajectory/intensity data?

Since both the LSTM encoder and decoder only accept one-dimensional data, we transform the 2D+T data into 1D+T features (using **red embedding network**) and similarly encode the 0 D+T data into 1D+T features (using **blue embedding network**). The two sets of features are then concatenated and fed into the encoder.

Question #14: Why isn't the red branch enough to take 2d data into account?

Answer #14: The red branch alone is not sufficient to fully account for the 3D data because it primarily focuses on extracting the spatial information from the 3D data and encoding it into feature vectors. While this step is crucial for capturing spatial dependencies, it does not handle the temporal aspect of the data, which is also critical in time-series tasks like TC forecasting.

The fusion of the red branch's spatial features with the 1D data is necessary but not complete until the combined features are processed by the LSTM to extract temporal dependencies. Without the contribution of the LSTM model, the information about how the data evolves over time would be missing. Thus, both branches are required to capture the complete picture: the red branch for spatial features and the LSTM for temporal features.

Question #15: How do you 'ensure that the De-LSTM input aligns with En-LSTM'? For me, the De-LSTM inputs is composed of En-LSTM outputs, and 2D-embedding outputs, no?

Answer #15: The alignment between the De-LSTM and En-LSTM inputs is ensured by keeping the input type of De-LSTM and En-LSTM the same. Autoregressive nature of LSTM is the reason why we need to align them. We are sorry that we have not explicitly highlighted this autoregressive

process. Now, we modify some subscription to explain it. As shown in Equations (5), at each iteration, the LSTM processes one-time-step's data and updates its internal states, continuously extracting temporal features from the historical TC data. The encoder follows this process for encoding.

Then, we add some subscriptions about the process of De-LSTM to detail explain **why we need to align them** and **how do we 'ensure that the De-LSTM input aligns with En-LSTM'** on pages 15-16, lines 1016-1057:

First, model fuses the features h_n and $e_{Env-time}$ by using an MLP ($\phi(\cdot)$) to obtain fused features h_{fused} , as shown in Equation (11). Then, h_{fused} and the noise vector z are concatenated and used as the initial status of Decoder. The model then selects a generator (g^i) by GC-Net and Monte Carlo method. $[h_{fused}, z]$ and the latest fused history data e_n^{En} are input into decoder function $De_{LSTM^i}(\cdot)$ of g^i to predict track and intensity of the TC \hat{Y}_{n+1} , as shown in Equation (12) ($t = 1$). We only predict 1D data, such as the track and intensity of TC, and do not predict 3D data. We believe that 3D data are more challenging to predict than 1D data, and the main task of our work is to predict the track and intensity of the TC, which are 1D data. Predicting 1D and 3D data simultaneously affects the performance of 1D data prediction. Therefore, we only predict the track and intensity of TC. However, if we want to predict the track and intensity after the $n + 1$ time step, we need to encode \hat{Y}_{n+1} using an MLP to obtain 1D features and fuse them with the 3D predicted features e_{3d}^{De} from Equation (2), as shown in Equations (13) and (14). We align the input types of En-LSTM and De-LSTM with both 1D and 3D features by the approach mentioned above. This method reduces the complexity of our model's prediction task by focusing primarily on 1D data prediction, while also providing the generator with additional information for more accurate predictions.

$$h_{fused} = \phi(h_n, e_{Env-time}; W_{h_f}) \quad (11)$$

$$\hat{Y}_{n+t}^i, h_{n+t}^i = \begin{cases} De_{LSTM^i}([h_{fused}, z^i], e_n^{En}; W_{LSTM_{De}^i}) & \text{if } t = 1, \\ De_{LSTM^i}(h_{n+t-1}^i, e_{n+t-1}^{De}; W_{LSTM_{De}^i}) & \text{others.} \end{cases} \quad (12)$$

$$e_{n+t}^{De_{1d}} = \phi(\hat{Y}_{n+t}^i; W_{e_{1d}^{De}}) \quad (13)$$

$$e_{n+t}^{De} = \phi(cat(e_{n+t}^{De_{1d}}, e_{n+t}^{De_{3d}}); W_{e^{De}}) \quad (14)$$

where i indicates that g^i is selected. h_{n+t}^i denotes the temporal-spatial features from the first $(n + t - 1)$ time steps of $Data_{1d}$ and $Data_{3d}$ of TC, $t \in \{1, 2, \dots, m\}$. m is the number of time steps for TC track and intensity that we want to predict. $e_{n+t}^{De_{1d}}$ and $e_{n+t}^{De_{3d}}$ are the encoded features of predicted 1D and 3D data at future time step t respectively. e_{n+t}^{De} represents the fused feature of $e_{n+t}^{De_{1d}}$ and $e_{n+t}^{De_{3d}}$, which is input into $De_{LSTM^i}(\cdot)$ to further predict the track and intensity of TC, as shown in Equation (12) (others). Equations (12), (13), and (14) are executed sequentially for m iterations to obtain the TC track and intensity prediction $\hat{Y}_i \equiv \{\hat{Y}_{n+1}^i, \hat{Y}_{n+2}^i, \dots, \hat{Y}_{n+m}^i\}$. W_{h_f} , $W_{LSTM_{De}^i}$, $W_{e_{1d}^{De}}$ and $W_{e^{De}}$ are the corresponding weights of

modules. We have now completed generating the future tendencies of the TC, concluding the generator phase. Next, we proceed to describe the discriminator phase.

Question #16: The 'Encoder 2d' in eq. 1 is not visible in the schema. Maybe consider adding sub-schemas to complement the methods section?

Answer #16: To clarify our model, we have revised the relevant descriptions and separated the explanation of the 'Encoder 3D (2D+t)' as shown in Equations (1) and (2). The framework figure has also been updated accordingly. Equation (1) corresponds to the 3D-UNet module in the red branch of the figure, while Equation (2) corresponds to the embedding module in the red branch. The corresponding content on page 14, lines 911-918 are as follow:

$$\hat{\mathbf{Y}}_{3d} = 3DUnet(\mathbf{X}_{3d}; W_{3D-Unet}) \quad (1)$$

$$e_{3d}^{En}, e_{3d}^{De} = \phi([\mathbf{X}_{3d}, \hat{\mathbf{Y}}_{3d}]; W_{MLP_{3d}}) \quad (2)$$

where \mathbf{X}_{3d} represents the history sequence $Data_{3d}$ of TC, $\hat{\mathbf{Y}}_{3d}$ represents the future predicted $Data_{3d}$ of TC, and e_{3d}^{En} represents the $Data_{3d}$ feature input for the 1D-Data Encoder, e_{3d}^{De} represents the predicted $Data_{3d}$ feature input for Generators. $W_{3D-Unet}$ and $W_{MLP_{3d}}$ represent the weights of the 3D-Unet and MLP respectively. Thus, the feature extraction of $Data_{3d}$ is completed, resulting in features e_{3d}^{En} and e_{3d}^{De} .

Question #17: Line 702. 'Then, e2d and e1d are concatenated as MLP input to produce fusion, which is fused with the features from two different dimensions'  what 2 dimensions are you talking about?

Answer #17: 'Two dimensions' is referring to the 3D and 1D data that have been encoded into 1D feature vectors for further processing. The LSTM can only accept one-dimensional inputs, so e_{3d}^{En} (we changed e_{2d}^{En} to e_{3d}^{En}) refers to the features extracted from the two-dimensional data with time series, which have been encoded into a one-dimensional feature vector. Meanwhile, e_{1d}^{En} refers to the features extracted from the zero-dimensional data (such as scalar values) with time series, which are also encoded into a one-dimensional feature vector. These two feature vectors, from different original dimensions (3D and 1D), are concatenated and input into a multilayer perceptron (MLP) to perform feature fusion. The fused feature vector is then fed into the LSTM to extract temporal information.

Question #18: Environmental module. The text lacks of clarity, in particular in order to understand the difference between this and the other networks, especially because you are using the data1d and data2d also..?

Answer #18: To clarify the distinction between the environmental module and other networks, we first note that, as shown in Table 2.3, the environmental data differs in both content and type from the 1D and 3D data. Therefore, it requires a new encoder for feature extraction. In the Environmental module (Env-T-Net), in addition to conventional CNN and MLP, we also used a Transformer [4] network for the environmental data. This network is capable of capturing both temporal relationships and interactions between different types of data, making it more effective for extracting environmental information. We also modify some description in the paper to make Environmental module clarity, which is shown in page 14, lines 939-954.

Unlike $Data_{1d}$ and $Data_{3d}$, the environmental data, as shown in Table 2.3, differs in both content and type, requiring a dedicated encoder for effective feature extraction. To address this, we designed an environmental module called the Environment-Time Network (Env-T-Net). In this module, besides conventional Convolutional Neural Networks (CNN) and Multi-Layer Perceptrons (MLP), we also used a Transformer (Vaswani et al. 2017) network for the environmental data. The CNN and MLP are tailored to accommodate environmental data across various modalities and dimensions. The Transformer module captures both temporal relationships and interactions between different types of environmental data. This approach enhances the module's effectiveness in extracting critical environmental information.

Question #19: Please indicate the size of the 2D windows, is it centered, not centered, and on what?

Answer #19: Sorry for that we do not provide enough information about it. We add some descriptions in page 5, lines 332-334 to explain it:

We crop the data covering $20^{\circ} \times 20^{\circ}$ region around the TC center. The spatial resolution of them is 0.25° and the time resolution is 6 hours.

Question #20: The 'monte-carlo roulette generator': any citations?

Answer #20: Sorry for that we did not give the citation about monte-carlo method. We have added it in this version manuscript. The Monte Carlo method is a classic sampling algorithm [5]. We apply this method to the selection of the generator. The main process is as follows: First, we obtain a set of probability values through GC-Net, with each probability representing the likelihood of the corresponding generator being selected. The sum of these probabilities equals 1. Using the Monte Carlo method, we sample each generator with a probability value six times. In each sampling round, generators with higher probabilities are more likely to be selected to predict TC data, but those with lower probabilities still have a chance to be chosen. We detail this process in section **Generator Chooser Network (GC-Net)** on page 15. This process is very similar to a roulette wheel, as shown in the figure.

Therefore, 'roulette' is a metaphor for the Monte Carlo method in our work. In summary, based on the existing probabilities of each generator, we use the Monte Carlo method for sampling, and we refer to this approach metaphorically as the roulette wheel.

Question #21: Are the different generators different? why? how does the discriminator help, could you not just perform a supervision on the values?

Answer #21: The different generators in our model share the same structure but do not share parameters. This means that the same input features, when passed through different generators, will produce different outputs.

Why are these generators different? The goal of training multiple generators with distinct parameters is to encourage each generator to develop a different prediction bias. For instance, the first generator may be more inclined to predict a TC moving in a southeastern direction. If the GC-Net uses historical and environmental data to determine that the TC is likely to move southeast, the probability of selecting the first generator will increase. This strategy makes it more likely that the model predicts the correct and reasonable trend.

How does the discriminator help, could we not just perform a supervision on the values?

In the context of a Generative Adversarial Network (GAN), the discriminator plays a crucial role in distinguishing between real data and the predicted data generated by the generator. More specifically, the discriminator is a binary classifier that outputs a probability indicating whether the input data is 'real' TC data (from the training dataset) or 'fake' TC data (generated by the generator). The discriminator's functions can be summarized as follows:

1. Providing feedback to the generator: The generator uses feedback from the discriminator to improve its output, gradually refining the generated data to better approximate the real data distribution. The generator's objective is to 'fool' the discriminator into thinking that the generated

data is real.

2. Enhancing the generator's quality: As the discriminator becomes more effective at distinguishing real data from fake data, it forces the generator to improve the quality of its outputs. This adversarial training leads to the generator producing highly realistic TC data over time.

3. Joint optimization: During GAN training, the generator and discriminator are in a competitive dynamic. The discriminator's loss is used to update its own parameters while also providing signals that help improve the generator's performance. This adversarial feedback loop ultimately improves the generator's ability to produce realistic TC data.

Without the discriminator (**if we just perform a supervision on the values**), the generator would lack a reliable feedback mechanism to evaluate the realism of its output. Relying solely on direct supervision of the generated values might not effectively align the generator's output distribution with the true TC data distribution, as there would be no adversarial feedback to push the generator to improve its predictions.

Question #22: 'TCNM was trained using a batch size of 96 over 100+q epochs, with the hyperparameter q set to 2'  ? why using a parameter q if it is just 102 in the end?

Answer #22: The parameter $q = 2$ was introduced to represent a specific training strategy aimed at improving the stability of the model. Specifically, during the first two epochs, we freeze the parameters of the generators and only update the parameters of the encoders and the GC-Net based on the GC-Net Loss. This allows the GC-Net to learn an initial capability of selecting the appropriate generator. By doing this, we ensure that the model has a more stable start to training, as the GC-Net becomes less likely to randomly select generators with equal probabilities. This warm-up period for the GC-Net helps the model focus on selecting the most suitable generator from the beginning. To make the description clearer, we modify 100+q to q+100 in the revision version.

If we were to start the entire training process immediately without this warm-up, the GC-Net might select generators randomly, which could undermine its ability to choose the best generator over time. Therefore, q serves to provide this warm-up phase, improving the overall performance and stability of the model.

Question #23: Please add a part about the format of the dataset, its functionality, etc. It is not explained and it not able to look at it for now as it is not released yet

Answer #23: We are sorry that we did not clarify the dataset TCN_D . Thus, we make a table to show the details of our dataset. 1D, 3D (2D+time) and environmental dataset, which is shown on page 7, Table 2.

For each variables' functionalities for deep learning methods' prediction, We add a new Table 3 on Page 8 to introduce them. Additionally, we have released the dataset along with the corresponding usage instructions on GitHub. To facilitate the review process, we provide both the full dataset and the separate test dataset for download. For more details, please visit:

<https://github.com/xiaochengfuhuo/TropiCycloneNet-Dataset>. A more detailed introduction will be updated on GitHub once this work is accepted.

Question #24: Typos

Answer #24: Thank you for helping us improve this paper. We modify the Typos that reviewers mentioned. Besides, we also check and correct Typos in the whole text. Some of the modifications are shown as follows:

1. The TCN_D ~~encompasses~~ includes near 70 years TC data.....
2. The TCN_D ~~comprises~~ not only includes the inherent attribute data of TC.....
3. According to Fig. 3, the dataset includes 3630.....
4. Some of ~~these~~ factors.....
5. First, ~~look~~ at these four methods:.....
6. Second, integrating M-GC, known for its advanced learning capabilities, significantly boosts ~~the~~ performance of ~~the~~ model.
7. While 3D-Unet is ~~widely~~ used in similar tasks, such as precipitation forecasting.
8. Overall, the ~~overall~~-training objective for the GAN component of TCN_M is summarized as follows:
9.denote the ~~weights~~ of 3D-Unet and MLP respectively.....

There is some references in this response letter:

- [1] Bi, K., Xie, L., Zhang, H. et al. Accurate medium-range global weather forecasting with 3D neural networks. Nature 619, 533–538 (2023). <https://doi.org/10.1038/s41586-023-06185-3>
- [2] Song, Hwanjun et al. "Learning From Noisy Labels With Deep Neural Networks: A Survey." IEEE Transactions on Neural Networks and Learning Systems 34 (2020): 8135-8153.
- [3] Liu, Z., Hao, K., Geng, X., Zou, Z., and Shi, Z. "Dual-Branched Spatio-Temporal Fusion Network for Multihorizon Tropical Cyclone Track Forecast." IEEE Journal of Selected Topics in Applied Earth Observations and Remote Sensing, vol. 15, 2022, pp. 3842–3852. DOI: 10.1109/JSTARS.2022.3170299.
- [4] Vaswani, A., Shazeer, N., Parmar, N., Uszkoreit, J., Jones, L., Gomez, A. N., Kaiser, Ł., Polosukhin, I. Attention is all you need. Advances in Neural Information Processing Systems. 30, (2017). https://proceedings.neurips.cc/paper_files/paper/2017/file/3f5ee243547dee91fbd053c1c4a845aa-Paper.pdf.
- [5] Metropolis, Nicholas, and S. Ulam. "The Monte Carlo Method." Journal of the American Statistical Association, vol. 44, no. 247, 1949, pp. 335–341. JSTOR, <https://doi.org/10.2307/2280232>.

Response to Comments from the Second Review Round:

Dear Reviewers:

We sincerely thank all reviewers for their valuable time and constructive feedback. We have carefully considered each comment and made substantial revisions accordingly, which is highlighted in blue font in the manuscript. In particular, we addressed concerns related to model input domain, comparisons with the large-scale weather forecasting model (e.g., Pangu-Weather) on long-term and extreme-case forecasting, dataset accessibility, and code clarity. We believe these revisions have significantly strengthened the manuscript in terms of clarity, completeness, and reproducibility. Detailed responses to each reviewer's comment are provided below.

For Reviewer #1

Question #1: However, as this method use input information from TC centered $20^\circ \times 20^\circ$ domain, predicting track with improved accuracy will always be restricted. As large-scale atmospheric flows influencing TC track are relatively stable, the authors could think of parameterizing it into their model domain for improving track performance.

Answer #1: Thank you for the reviewer's comments. It is indeed true that the TC-centered $20^\circ \times 20^\circ$ domain lacks information on large-scale atmospheric flows. However, we do consider this issue during our initial model design and data selection. Our decision to use the $20^\circ \times 20^\circ$ domain was based on the following reasons:

- (1) **Computational efficiency and local feature extraction:** Processing high-resolution gridded data (e.g., ERA5) over a large area significantly increases computational costs. The $20^\circ \times 20^\circ$ window allows us to effectively capture key features of the tropical cyclone core structure while maintaining reasonable computational efficiency. This setup ensures that the model consistently focuses on the most relevant TC information. Additionally, using a $20^\circ \times 20^\circ$ window helps our model avoid information redundancy that may arise from excessively large domains. For instance, in Pangu-Weather, which uses global-scale data as input (large domains), the model tends to underestimate TC intensity. This issue may stem from

redundant information in the input data, causing the model to overlook crucial features within the TC center.

- (2) **Supplementation with environmental data:** The characteristics of the TC itself differ from the environmental factors influencing it. To account for this distinction, we designed our model to decouple the feature extraction processes for these two types of data. Specifically, we introduced separate modules: the 3D-Data Encoder for the TC's core structure and the Env-T-Net for large-scale environmental information. As a result, we set the spatial extent for the 3D data to $20^{\circ} \times 20^{\circ}$ to ensure that the 3D-Data Encoder focuses solely on TC-specific features. Meanwhile, to compensate for the missing large-scale environmental information in the $20^{\circ} \times 20^{\circ}$ domain, we incorporated environmental data into the TCN_D dataset and designed the Env-T-Net module to extract these large-scale environmental factors.
- (3) **Model design adaptability:** The reviewer suggested using a variable domain size as a parameter. However, our deep learning model requires a fixed input size and cannot dynamically adjust the spatial dimensions of the input data. Nevertheless, we find the reviewer's suggestion insightful. Inspired by this, we plan to explore an approach where we input environmental fields at multiple spatial scales within our computational capacity and design an attention mechanism that allows the model to adaptively extract relevant information from different scales. This strategy would effectively mitigate information redundancy while further decoupling the feature extraction processes for TC-specific and large-scale environmental features.

Question #2: The README file provide necessary instructions for installing and running the application. I configured and installed the code but did not run it using datasets provided in google drive.

Answer #2:

Thank you for acknowledging the documentation provided with our code. We have provided two types of datasets:

Raw dataset: This dataset is stored in NetCDF (nc) format, which helps reduce storage space and allows future researchers to process the data more flexibly. This dataset has been uploaded to Zenodo, and the download link is as follows: <https://doi.org/10.5281/zenodo.15009527>

Processed dataset for TCN_M model testing: This dataset is stored in NumPy (npz) format, which is more convenient for direct testing with the TCN_M model. Due to its relatively large storage requirements, we have currently provided only the data from 2018-2019 for review purposes, which is shown at <https://github.com/xiaochengfuhuo/TropiCycloneNet>. The TCN_M code files have also been uploaded to Zenodo, with the following link: <https://doi.org/10.5281/zenodo.15024028>

For Reviewer #2

Question #1: the fact that the trajectory forecast results of new Weather forecast models are substantially better than this paper's results (new figure 5). The long-term forecast figure

(5b) does not contain them, but I guess the deep learning weather forecast would also surpass the other methods. The intensity forecast is thus the added value, but it makes the paper weaker.

Answer #1: We would like to address the comparison between our model and large-scale weather models. While we understand the concern regarding the trajectory forecast results, it is important to note that such comparisons are not entirely fair due to the substantial differences in training resources and costs. For example, models like Pangu-Weather require 192 V100 GPUs and 64 days of training, while our model is trained with a single A6000 GPU in just one day. These computational requirements highlight the practicality of our approach in terms of accessibility, especially for communities who may not have access to large-scale computing infrastructure.

Additionally, in the previous version, Reviewer #1 suggested evaluating the model's performance in long-term forecasts and during rapid changes in TC track or intensity. However, since the reviewer did not specifically mention comparison with large models, we initially overlooked including such comparisons in these two scenarios. In this revision, we have added a comparison with the large-scale model Pangu-Weather. The results are shown in Fig.5. b and c:

Figure 5: Comparisons of TCN_M with other methods across various aspects. The comparative results are based on WP sea area data, as most of the comparative methods only provide performance results for the WP region. Sub-figures (a-1), (a-2), and (a-3) present the average absolute error comparisons of TC predictions among different deep learning methods. Sub-figures (b-1), (b-2), and (b-3) show the long-term forecasting (6-72 h) capabilities of TCN_M . Sub-figures (c-1), (c-2), and (c-3) demonstrate the performance of TCN_M during rapid changes in TC track or intensity compared to its overall performance on the entire dataset. In these sub-figures, The three solid lines in the represent the results of CMO, Pangu-Weather, and TCN_M on the complete test set, respectively. $CMO-r_{Int}$, $Pangu-r_{Int}$, and TCN_M-r_{Int} represent the results of CMO, Pangu-Weather, and TCN_M on rapid intensifying TC cases, respectively. $CMO-r_{Tra}$, $Pangu-r_{Tra}$, and TCN_M-r_{Tra} represent the results of CMO, Pangu-Weather, and TCN_M on curving TC cases, respectively. Best viewed in color.

It can be observed that Pangu, a model designed for global weather variable forecasting, does not perform perfectly in tropical cyclone prediction. First, in the long-term forecast scenario (b-1), Pangu achieves good performance in TC track prediction, but the performance of Pangu in intensity prediction is still the worst. Second, in the rapid changes in TC track or intensity scenario (c-1), Pangu's overall track forecasting performance deteriorates significantly under extreme conditions compared to our model. In particular, its performance in sharply recurving cases is noticeably worse. These results suggest that large-scale models designed for global forecasting are less capable of capturing extreme variations in tropical cyclone systems, whereas our model, specifically designed for TC prediction, demonstrates superior performance in such challenging scenarios. We have added descriptions and explanations of this phenomenon on page 12, lines 735-748:

As shown in Fig.5.c-1 and c-2, we also show the performance of Pangu-Weather on rapid intensifying and recurving TC Cases. Experimental results demonstrate that under extreme conditions, the performance of large-scale model Pangu-Weather is inferior to TCN_M in both track and intensity forecasting. In particular, Pangu-Weather shows a significant drop in track forecasting accuracy when dealing with recurving TC cases. This indicates that, as a global forecasting model, Pangu-Weather struggles to accurately capture certain extreme variations in TC systems. In contrast, our model is specifically designed for tropical cyclone prediction and exhibits superior performance under such extreme scenarios. These findings underscore the importance of the method proposed in this paper.

Besides, while it is true that those large weather forecasting models outperform our model in overall track forecasting capability, our model excels in predicting tropical cyclone intensity, a task that remains a significant challenge for the large models (a-2, b-2, and c-2). As we are aware, many of the large-scale models (such as Pangu-Weather, Feng Wu, GraphCast) struggle with intensity forecasting. Our model's better performance in this area demonstrates its potential for further advancing TC prediction and can serve as a valuable reference for future model development.

What is more, we also wish to highlight the role of our proposed method, as an open-source baseline. One of the key motivations behind our work is to provide a well-performing and accessible baseline for researchers interested in TC prediction. Many large models, including Pangu-Weather, Feng Wu, and GraphCast, do not fully open-source their training code, making it difficult for others to build on these models. By contrast, our method is fully open-source, allowing researchers to quickly get started with TC forecasting without needing extensive computational resources. The accessibility of our model in terms of computational cost (requiring only a single A6000 GPU for training) further facilitates its adoption by the research community and encourages the development of new, innovative methods in this area.

Question #2 : The database, now clearer and accessible, is still not a proper benchmark because the dataset is stored on a google drive and not on a repository with a DOI and thus permanent link (such as Zenodo), and the authors want to keep updating it. I see the value of it, of course, but it means that it cannot be used for a long-term benchmark as it could be removed/modified. As the database is made of open-source fields, not novel by themselves

except the environmental fields, the added value of the repository is only if it can become a new solid benchmark dataset.

Answer #2: We sincerely appreciate Reviewer #2's recognition of the value of our dataset. Indeed, a significant contribution of our work is the creation of the first high-quality global multimodal TC forecasting dataset.

To ensure long-term accessibility and usability as a benchmark, we have now uploaded the complete dataset to Zenodo, which provides a DOI and a permanent link. The dataset can be accessed at: <https://doi.org/10.5281/zenodo.15009527>. This will allow it to serve as a solid benchmark for future research in the field.

Additionally, we are committed to continuously updating the dataset. Any updates will be synchronized on both our GitHub (<https://github.com/xiaochengfuhuo/TropiCycloneNet-Dataset>) repository and Zenodo (by creating new project versions). By providing and maintaining this dataset, we aim to foster greater participation from the research community, which is crucial for advancing TC prediction research.

Question #3: I just checked quickly the repository, it looks correct with some explanation on how to use it. Yet all comments are for now in chinese only. The authors said that they would clean the code afterwards

Answer #3: Thank you for acknowledging the correctness of our provided code and documentation. We have now cleaned the code and added the necessary English comments for better clarity and accessibility. Additionally, we have uploaded the updated version of our code to Zenodo, ensuring long-term availability. The repository can be accessed at: <https://doi.org/10.5281/zenodo.15024028>. We appreciate your feedback and hope this improvement enhances usability for a broader research community.